



# Carbon isotopes in the marine biogeochemistry model FESOM2.1-REcoM3

Martin Butzin[1,2], Ying Ye[1], Christoph Völker[1], Özgür Gürses[1], Judith Hauck[1], Peter Köhler[1]

[1]Alfred-Wegener-Institut Helmholtz-Zentrum für Polar- und Meeresforschung, D-27515 Bremerhaven, Germany
[2]Now at MARUM-Center for Marine Environmental Sciences, University of Bremen, D-28334 Bremen, Germany

*Correspondence to*: Martin Butzin (mbutzin@marum.de)

**Abstract.** In this paper we describe the implementation of the carbon isotopes $^{13}C$ and $^{14}C$ (radiocarbon) into the marine biogeochemistry model REcoM3. The implementation is tested in long-term equilibrium simulations where REcoM3 is coupled with the ocean general circulation model FESOM2.1, applying a low-resolution configuration and idealized climate forcing. Focusing on the carbon-isotopic composition of dissolved inorganic carbon ($\delta^{13}C_{DIC}$ and $\Delta^{14}C_{DIC}$), our model results are largely consistent with reconstructions for the pre-anthropogenic period. Our simulations also exhibit discrepancies, e.g., in upwelling regions and the interior of the North Pacific. Some of these differences are due to the limitations of our ocean circulation model setup which results in a rather shallow meridional overturning circulation. We additionally study the accuracy of two simplified modelling approaches for dissolved inorganic $^{14}C$, which are faster (15 % and about a factor of five, respectively) than the complete consideration of the marine radiocarbon cycle. The accuracy of both simplified approaches is better than 5 % which should be sufficient for most studies of $\Delta^{14}C_{DIC}$.

## 1 Introduction

Carbon isotopes are powerful tools for tracing present and past biogeochemical cycles and water masses. The stable isotope carbon-13 ($^{13}C$) can be used to study the anthropogenic perturbation of the global carbon cycle due to the combustion of isotopically depleted fossil fuels via the so-called $^{13}C$ Suess effect (e.g., Keeling 1979; Quay et al., 1992; Köhler 2016; Graven et al., 2021). Carbon-13 may also help to decipher the exchange between atmospheric, marine and terrestrial carbon reservoirs in the past, for example during the last glacial cycle (Köhler et al., 2006, Ciais et al., 2012, Broecker and McGee, 2013, Jeltsch-Thömmes et al., 2019, Menking et al., 2022). Furthermore, $^{13}C$ is a proxy for oceanic nutrients and can be employed to infer past marine biological productivity, export production, and water mass distributions assuming that calcareous tests of foraminifera are faithful recorders of dissolved inorganic $^{13}C$ (e.g., Broecker and Peng, 1982; Lynch-Stieglitz et al., 2007; Hesse et al., 2011; Schmitcher et al. 2017). The radioactive isotope carbon-14 (radiocarbon, $^{14}C$) is the most important geochemical



chronometer for dating organic matter and for assessing ocean ventilation rates and pathways over the
35  last 55000 years (Heaton et al., 2021; Rafter et al., 2022; Skinner and Bard, 2022, Skinner et al., 2023).
In addition, the penetration of bomb-produced $^{14}$C into the oceans has provided a benchmark for ocean
circulation models (Matsumoto et al., 2004). During the last decade, numerous ocean general circulation
models have been equipped with carbon isotopes and applied in Earth system modelling studies (e.g.,
Tschumi et al., 2011; Holden et al., 2013; Schmittner et al., 2013; Jahn et al., 2015; Menviel et al., 2015;
40  Buchanan et al., 2019; Jeltsch-Thömmes et al., 2019; Dentith et al., 2020; Tjiputra et al., 2020; Claret et
al., 2021; Liu et al. 2021; Morée et al., 2021).

Here, we describe the implementation of both carbon isotopes into the marine biogeochemistry model
REcoM3 which is part of the AWI Earth System Model. The technical details will be described in Section
2 and simulations in Section 3. Isotope results will be presented in terms of $\delta^{13}$C and $\Delta^{14}$C which express
the relative deviations of observed $^{13}$C/$^{12}$C ($^{13}R$) and $^{14}$C/$^{12}$C ($^{14}R$) ratios with respect to specific standard
values ($^{13}R_{std}$ = 0.0112372; Craig, 1957, and $^{14}R_{std}$ = 1.176 × 10$^{-12}$; Karlén et al., 1964), where

$$\delta^{13}C = [^{13}R\ /\ ^{13}R_{std} - 1] \cdot 1000, \tag{1}$$

$$\delta^{14}C = [^{14}R\ /\ ^{14}R_{std} - 1] \cdot 1000, \tag{2}$$

and (following Stuiver and Pollach, 1977)

$$\Delta^{14}C = \delta^{14}C - 2\ (\delta^{13}C + 25\ ‰)\ (1 + \delta^{14}C\ /\ 1000). \tag{3}$$

Section 4 will conclude with a brief summary.

## 2 Model description

### 2.1 Short overview of REcoM3

The Regulated Ecosystem Model, version 3 (REcoM3) is described in detail by Gürses et al. (2023). Here,
we will only give a brief summary of the common model features and describe the differences to the
configuration that we use in this study. REcoM considers the marine biogeochemical cycles of carbon,
nitrogen, silicon, iron, and oxygen. The ecosystem component of REcoM includes nutrients, two
phytoplankton functional types (distinguishing between small phytoplankton and diatoms), one
zooplankton functional type, one detritus type, and dissolved organic matter. Different to most other
marine biogeochemistry models, REcoM does not rely on a fixed internal stoichiometry of phytoplankton.
Instead, the composition of organic soft tissue is regulated (i.e., calculated) in response to light,
temperature and nutrient supply which enables to assess stoichiometric shifts between present and past.
The model includes a sediment layer in which sinking detritus (consisting of particulate organic matter,
calcite and opal) is fully remineralized and where solutes are returned to the bottom water layer.
Alternatively, REcoM3 can be run with the comprehensive sediment model MEDUSA2 (Munhoven,
2021) which will be described in another paper (Ye et al., in prep.). Apart from the implementation of





carbon isotopes, the main difference to the standard REcoM3 configuration by Gürses et al. (2023) is that REcoM3 considers two zooplankton and two detritus groups instead of one, which would require to include six additional carbon-isotopic tracers at the expense of model speed. We refer to the configuration

presented here as REcoM3p as an initial set-up for paleo studies. To simulate biogeochemical tracer circulation, REcoM3 needs a transport model. Here, we utilize the ocean general circulation model FESOM2.1 which is an update of FESOM2.0 (Danilov et al., 2017). The coupling of FESOM2.1 with REcoM3 is also described by Gürses et al. (2023), where all biogeochemical model equations except for carbon isotopes can be found.


## 2.2 Implementation of carbon isotopes

Carbon-13 and $^{14}$C are implemented as additional passive tracers, tripling the number of carbon-containing tracers in REcoM from 8 to 24. In the kinetic calculations of the carbonate system $^{13}$C and $^{14}$C are neglected because their abundances are small. For the same reason, and to ensure numerical stability

in the other model calculations involving $^{13}$C and $^{14}$C, we do not apply the true values of the isotopic standard ratios but set $^{13}R_{std} = 1$ and $^{14}R_{std} = 1$ following Jahn et al. (2015). As a consequence, $^{12}$C, $^{13}$C, and $^{14}$C concentrations are of the same magnitude but the scaling factors cancel out when $^{13}$C and $^{14}$C concentrations are converted to $\delta^{13}$C and $\Delta^{14}$C values. We consider isotopic fractionation during air-sea gas exchange, dissolution of $CO_2$ in seawater, and photosynthesis by phytoplankton. In addition, the

model accounts for radioactive decay and, optionally, cosmogenic production of $^{14}$C. The details are explained in the following subsections.

## 2.3 Carbon-13

### 2.3.1 Air-sea exchange

The isotopic fractionation during uptake and dissolution of $^{13}CO_2$ is calculated according to Zhang et al. (1995), mostly following the biogeochemical protocol of the CMIP6 Ocean Model Intercomparison Project (OMIP-BGC protocol, Orr et al., 2017). That is, the air-sea exchange flux $^{13}F$ is proportional to the difference between the saturation and in-situ concentrations of aqueous $^{13}CO_2$:

$$^{13}F \quad = k_w \left( [^{13}CO_2{}^*]_{sat} - [^{13}CO_2{}^*] \right)$$

$$= k_w \left( {}^{13}\alpha_k \, {}^{13}\alpha_{aq-g} - 0.0002 \right) \left( {}^{13}R_{atm} [CO_2{}^*]_{sat} - {}^{13}R_{DIC} / {}^{13}\alpha_{DIC-g} [CO_2{}^*] \right), \quad (4)$$

where $k_w$ is the $CO_2$ gas transfer velocity (calculated according to Wanninkhof, 2014, additionally considering sea-ice cover), and $[^{13}CO_2{}^*]_{sat}$ and $[^{13}CO_2{}^*]$ are the saturation and in-situ concentrations of aqueous $^{13}CO_2$. $^{13}R_{atm}$ and $^{13}R_{DIC}$ are the $^{13}$C/$^{12}$C concentration ratios of atmospheric $CO_2$ and dissolved inorganic carbon (DIC). Isotopic fractionation factors $\alpha$ denote kinetic fractionation during $CO_2$ gas



transfer ($^{13}\alpha_k$), equilibrium fractionation during gas dissolution ($^{13}\alpha_{aq-g}$), and equilibrium fractionation between DIC and gaseous $CO_2$ ($^{13}\alpha_{DIC-g}$). Numerical values were taken from Zhang et al. (1995) who measured kinetic and equilibrium fractionation of $^{13}C$ in acidified freshwater. For kinetic fractionation we employ a constant value of $^{13}\alpha_k = 0.99912$ which is the average between 5 °C ($^{13}\alpha_k = 0.99919$) and 21 °C ($^{13}\alpha_k = 0.99905$). Equilibrium fractionation between aqueous and atmospheric $^{13}CO_2$ is expressed as

$$^{13}\alpha_{aq-g} = 1 + 0.001 \, (0.0049 \, T - 1.31) \tag{5}$$

where $T$ (°C) is the temperature of surface water. The numerical values of $^{13}\alpha_k$ and $^{13}\alpha_{aq-g}$ were determined for fresh water. To account for enhanced $^{13}C$ fractionation in seawater associated with hydration reactions, equation (4) includes a correction factor of -0.0002 following Zhang et al. (1995) which is not considered in the OMIP-BGC protocol. Fractionation between DIC and gaseous $CO_2$ is calculated as

$$^{13}\alpha_{DIC-g} = 1 + 0.001 \, ((0.014 \, f\text{CO}_3 - 0.107) \, T + 10.53), \tag{6}$$

where $f\text{CO}_3 = [\text{CO}_3^{2-}] / \text{DIC}$ is the carbonate fraction of DIC.

### 2.3.2 Biogenic fractionation

Photosynthesis of phytoplankton leads to isotopic depletion of particulate organic carbon (POC) which is expressed following Freeman and Hayes (1992):

$$[^{13}\text{C}_{POC}] = {}^{13}R_{POC} \, [^{12}\text{C}_{POC}] = {}^{13}R_{aq} / {}^{13}\alpha_p \, [^{12}\text{C}_{POC}] \tag{7}$$

where $[^{13}\text{C}_{POC}]$ and $[^{12}\text{C}_{POC}]$ are the isotopic POC concentrations in phytoplankton, $^{13}R_{POC}$ is the corresponding isotopic ratio, $^{13}R_{aq}$ is the $^{13}C/^{12}C$ concentration ratio of aqueous $CO_2$, and $^{13}\alpha_p$ is the isotopic fractionation factor associated with photosynthesis. Various experimental and modelling studies have determined $^{13}\alpha_p$ for certain phytoplankton species (Laws et al., 1995; Rau et al., 1996; Bidigare et al., 1997; Laws et al., 1997; Rau et al., 1997; Popp et al., 1998; Keller and Morel, 1999). However, it is uncertain to what extent these studies are globally representative and can be transferred into a single global modelling framework (see also the review by Brandenburg et al., 2022). Therefore, we pursue a less sophisticated but robust approach to calculate $^{13}\alpha_p$ which has been inferred from a global dataset of 525 $\delta^{13}\text{C}_{POC}$ field measurements spanning the period 1962–2010 CE (Young et al., 2013):

$$^{13}\alpha_p = 1 + 0.001 \, (17.6 \, (1 - 2.02 / [\text{CO}_2^*])), \tag{8}$$

where $[\text{CO}_2^*]$ is in μmol L$^{-1}$. Since no distinction is made between different phytoplankton species in equation (8), the carbon-isotopic composition of small phytoplankton and diatoms in our model is the same. Similarly to other models (e.g., Schmittner et al., 2013; Menviel et al., 2105; Buchanan et al., 2019; Tjiputra et al., 2020; Liu et al., 2021) we do not consider carbon-isotopic fractionation during formation and dissolution of biogenic calcite because the effect is small and varies between species ($\alpha \sim 0.999–1.003$ according to Ziveri et al., 2003).

### 2.4 Radiocarbon

Radiocarbon is subject to radioactive decay, cosmogenic production, and isotopic fractionation. The radioactive decay (applying a half-life of 5700 years, Audi et al., 2003; Bé and Chechev, 2012; Kutschera, 2013) is balanced in the model by cosmogenic $^{14}C$ production fluxes or, alternatively, by prescribed atmospheric $^{14}CO_2$ concentrations corresponding to atmospheric $\Delta^{14}C$ values.

Fractionation factors are calculated analogously to $^{13}\alpha$, with $^{14}\alpha = 2\ ^{13}\alpha - 1$ (e.g., Craig, 1954). We have implemented $^{14}C$ in two ways. The first approach ("CC") considers the complete $^{14}C$ cycle parallel to $^{13}C$. The second, approximate approach ("IC") disregards isotopic fractionation of $^{14}C$ and radioactive decay of organic matter. In turn, $DI^{14}C$ and DIC have identical sources and sinks except for atmospheric $CO_2$ and radioactive decay. This "inorganic" $^{14}C$ approach is conceptually similar, but not identical, to the "abiotic" $^{14}C$ modelling approach described in the OMIP biogeochemical protocol (Orr et al., 2017). In our IC approach, DIC and $DI^{14}C$ include biological sources and sinks. This does not apply to the OMIP-abiotic approach, which also considers alkalinity in a simplified way. In Section 3.3 we will scrutinize the accuracy of the IC approximation.

It has been shown that marine $\Delta^{14}C$ values of DIC ($\Delta^{14}C_{DIC}$) are primarily governed by transport and radioactive decay (Fiadeiro, 1982; see also Mouchet, 2013). This implies that $\Delta^{14}C_{DIC}$ can be implemented into ocean general circulation models without a full carbon cycle model (cf. Toggweiler et al., 1989, and numerous other studies later on). We evaluate this $\Delta^{14}C$ approximation ("DA") in an additional simulation and compare it with REcoM approaches CC and IC also in Section 3.3.

A list of the various model experiments and their key features can be found in Table 1.

## 3 Results and discussions

### 3.1 Simulated ocean climatology

FESOM employs unstructured meshes with variable horizontal resolution. The default mesh of FESOM2.1-REcoM3 has about 127000 horizontal surface nodes (Gürses et al., 2023). Here, our model resolution is radically reduced, considering 3140 surface nodes corresponding to a median horizontal resolution of 260 km (the so-called pi-mesh, see Fig. A1 in Appendix A). This configuration requires fewer computational resources and allows to perform simulations over the time scale of marine carbon isotope equilibration (i.e. over several thousand years) within a few weeks. Vertical resolution is 47 layers using $z^*$ coordinates with nonlinear free surface (further details see Scholz et al., 2019).

In a first step we run FESOM without REcoM over 1000 years to spinup the overturning circulation and thermohaline fields. FESOM was initialized with seasonal winter temperatures and salinities by Steele et al. (2001), and driven with annually repeated atmospheric fields using Corrected Normal Year Forcing Version 2.0 (Large and Yeager, 2009; for an overview see also Griffies et al., 2012). As FESOM2.1 had



originally been adjusted for higher resolution and different forcing, we retuned the model in our setup by reducing the maximum Gent-McWilliams thickness diffusivity from originally 2000 m² s⁻¹ to 1000 m² s⁻¹. After 1000 simulated years there was no significant drift of thermohaline and circulation fields, REcoM3p was turned on, and both models were run over additional 5000 years. REcoM3p was initialized with gridded concentration fields of total alkalinity, DIC and nitrate from GLODAPv2 (Key et al., 2015; Lauvset et al., 2016), oxygen and silicate from WOA13 (Garcia et al., 2014a, 2014b), and dissolved iron according to Aumont et al. (2003) and Tagliabue et al. (2012). Dissolved inorganic $^{13}C$ and $^{14}C$ were initialized with DIC, assuming initial fractionation values of $\delta^{13}C_{DIC} = 0$ ‰ and $\Delta^{14}C_{DIC} = -150$ ‰, respectively. REcoM3p was forced with constant atmospheric $CO_2$ concentrations ($[^{12}CO_2]_{atm} = 284.3$ ppmv; $[^{13}CO_2]_{atm}$ and $[^{14}CO_2]_{atm}$ corresponding to $\delta^{13}C_{atm} = -6.61$ ‰ and $\Delta^{14}C_{atm} = 0$ ‰, respectively, following Orr et al., 2017), and with climatological-mean monthly fluxes of aeolian iron deposition (Albani et al., 2014).

As discussed in the following, our low-resolution test setup sufficiently captures the basic thermohaline circulation structures obtained with FESOM2.0 in higher-resolution simulations (Scholz et al., 2019, 2022). Compared to observations, our simulations exhibit a warm bias for thermocline and intermediate water as well as for North Atlantic Deep Water (NADW; see Fig. A2 in Appendix A). The most notable exception is the region of the North Atlantic Gulf stream which is not properly resolved and where upper ocean temperatures are considerably underestimated. The temperature biases covary with salinity biases. That is, simulated salinities are also higher than observations where simulated temperatures are high, and salinities are too low where simulated temperatures tend to be low (Fig. A3). In the Atlantic, FESOM arrives at a maximum meridional overturning circulation (AMOC) of about 16 Sv (1 Sv = 1 × 10⁶ m³ s⁻¹, Fig. A4), which is at the lower bound of observations while the simulated overturning cell is too shallow compared to observational estimates (see Buckley and Marshall, 2016; and further references therein). For the purposes of this study, our simulations also reasonably reflect the observed large-scale pattern of DIC (Key et al., 2015; Lauvset et al., 2016). That is, low concentrations are found at upper levels of the subtropical oceans and in the freshly ventilated interior of the North Atlantic, while DIC concentrations progressively increase in the South Atlantic and in the deep North Pacific (Fig. A5).

## 3.2 Carbon-13

We now focus on the carbon-isotopic composition of DIC near the sea surface and along meridional sections through the Atlantic and Pacific. Regarding $\delta^{13}C_{DIC}$, we compare our model results with the gridded preindustrial (PI) $\delta^{13}C_{DIC}$ climatology by Eide et al. (2017). Their reconstruction does not consider the upper 200 m to exclude the $^{13}C$ Suess effect.

In wide areas, our simulated near-surface $\delta^{13}C_{DIC}$ values are in the range of 1 to 2 ‰ (Fig. 1a). Higher $\delta^{13}C_{DIC}$ values are simulated in the subtropical oceans, particularly in the Atlantic, southeast Pacific and southern Indian Ocean. Isotopic depletion of up to ~ -1 ‰ is found in the eastern equatorial Pacific, the





subpolar North Pacific, the Bay of Bengal, and in the Angola Basin. While our simulation captures the reconstructed spatial pattern (shown in Fig. 1b), the simulated range of $\delta^{13}C_{DIC}$ variations is larger than in the reconstruction. That is, the model results are too high in the lower latitudes and too low in the above

mentioned depletion regions (Fig. 2a).

Considering the interior of the Atlantic Ocean, our simulation yields higher $\delta^{13}C_{DIC}$ in the North Atlantic than in the South Atlantic, and small vertical $\delta^{13}C_{DIC}$ gradients in the high latitudes of both hemispheres (Fig. 1c). In the Pacific, our simulation displays a reversed meridional $\delta^{13}C_{DIC}$ gradient with the strongest isotopic depletion at intermediate depths in the North Pacific. Our results are roughly in line with the

reconstruction by Eide et al. (2017, cf. Fig. 1d) but overestimate $\delta^{13}C_{DIC}$ in the upper Pacific at low latitudes as well as in the North Pacific between 1.5 km and 3 km depth (Fig. 2b). Similar inaccuracies are seen in other models (Tagliabue an Bopp, 2008; Schmittner et al., 2013; Jahn et al., 2015; Buchanan et al., 2019; Jeltsch-Thömmes et al., 2019; Dentith et al., 2020; Tjiputra et al., 2020; Liu et al., 2021). To some extent, the differences between simulated and observed $\delta^{13}C_{DIC}$ ($\Delta\delta^{13}C_{DIC}$) can be attributed by

decomposing $\delta^{13}C_{DIC}$ into biologic and thermodynamic sources,

$$\delta^{13}C_{DIC} = \delta^{13}C_{BIO} + \delta^{13}C_{AS}, \tag{9}$$

where $\delta^{13}C_{BIO}$ specifies the imprint of isotopic fractionation, respiration and remineralization of organic matter in the absence of air-sea exchange. Following Broecker and Maier-Reimer (1992), $\delta^{13}C_{BIO}$ is frequently determined from the covariation of $\delta^{13}C_{DIC}$ with marine phosphate ($PO_4^{3-}$). We adopt this

approach, employing revised parameter values by Eide et al. (2017) and tentatively replacing $PO_4^{3-}$ with dissolved inorganic nitrogen (DIN) divided by 16 because $PO_4^{3-}$ is not considered by REcoM3p,

$$\delta^{13}C_{BIO} = (2.8 - 1.1\ PO_4^{3-})\ ‰ \equiv (2.8 - 0.069\ DIN)\ ‰, \tag{10}$$

where $PO_4^{3-}$ and DIN are in $\mu mol\ kg^{-1}$. The thermodynamic component $\delta^{13}C_{AS}$ describes the effects of air-sea exchange and ocean circulation and is the residual of observed $\delta^{13}C_{DIC}$ and reconstructed $\delta^{13}C_{BIO}$.

In the following we compare $\Delta\delta^{13}C_{DIC}$ with the differences between the simulated and reconstructed $\delta^{13}C_{DIC}$ components, $\Delta\delta^{13}C_{BIO}$ and $\Delta\delta^{13}C_{AS}$. It appears that $\Delta\delta^{13}C_{DIC}$ corresponds to $\Delta\delta^{13}C_{BIO}$ in the low latitudes, upwelling regions, and the interior of the Atlantic (cf. Figs. 2 and 3), which points to model deficiencies in describing the sinking and regeneration of $^{13}C$-depleted organic carbon. On the other hand, $\Delta\delta^{13}C_{DIC}$ corresponds to $\Delta\delta^{13}C_{AS}$ in the upper thermocline of the open oceans in the Southern hemisphere

(shown in Fig. 4). While our results may generally suffer from the coarse model resolution and simplified climate forcing, the specific reasons for this correspondence are not obvious. As a residual term $\delta^{13}C_{AS}$ may also reflect effects of biological $^{13}C$ cycling which are not captured by Equation (10). For example, $\delta^{13}C_{BIO}$ is estimated for constant isotopic fractionation of marine organic matter of -19 ‰ while $\delta^{13}C_{POC}$ varies by about 10 ‰ according to field data (Verwega et al., 2021; see also Fig. 6a).



Therefore, we explore the sensitivity of $\delta^{13}C_{DIC}$ to biogenic fractionation in an additional experiment ("NP") in which biogenic fractionation is disabled (i.e., $^{13}\alpha_p = 1$ in equation (8)). Compared to the default simulation, $\delta^{13}C_{DIC|NP}$ decreases by up to 1 ‰ in the pelagic euphotic zone while $\delta^{13}C_{DIC|NP}$ increases in the disphotic zone below, which is particularly the case in highly productive regions (Fig. 5a-b). In the aphotic interior of the ocean $\delta^{13}C_{DIC|NP}$ progressively increases from the North Atlantic towards the North

Pacific by up to 2.4 ‰ (Fig. 5c). These findings are in line with similar sensitivity studies (Schmittner et al., 2013; Dentith et al., 2020) as well as with simulations comparing different parametrizations for $^{13}\alpha_p$ (Jahn et al., 2015; Buchanan et al., 2019; Dentith et al., 2020; Liu et al., 2021).

Our default simulation with enabled photosynthetic fractionation yields $\delta^{13}C_{POC}$ values between -18 and -25 ‰ while observations from the last decades range from -15 to -35 ‰ (Verwega et al., 2021; Fig. 6a).

The model overestimates $\delta^{13}C_{POC}$ at most locations (Fig. 6b). According to sensitivity experiment NP, the overestimation of $\delta^{13}C_{POC}$ should result in overly enriched $\delta^{13}C_{DIC}$ in the twilight and dark zones of highly productive regions (Fig. 5b) while Figure 2a indicates that the opposite is the case. However, this is only an apparent contradiction because the $\delta^{13}C_{POC}$ observations are biased by the $^{13}C$ Suess effect. Moreover, they exhibit a negative trend (by -3 ‰ between 1960 and 2010) which is about twice as high

as the known $^{13}C$ Suess effect on aqueous $CO_2$ (Young et al., 2013; Verwega et al., 2021). It has been presumed that this trend also reflects a shift in phytoplankton species composition (Lorrain et al., 2020; Verwega et al., 2021). Both effects are not considered in our simulation. A conclusive analysis would require transient simulations including historical values of atmospheric $^{13}CO_2$.

### 3.3 Radiocarbon

We consider $\Delta^{14}C_{DIC}$ and compare our model results with gridded fields of pre-bomb $\Delta^{14}C_{DIC}$ provided by the Global Ocean Data Analysis Project (GLODAPv1.1, Key et al., 2004).

In the comprehensive radiocarbon cycle simulation (CC) $\Delta^{14}C_{DIC|CC}$ is in the range of -40 to -140 ‰ (average value: -65 ‰) near the surface (at 50 m), with the highest values in the subtropical gyres and the lowest values in the Southern Ocean (Fig. 7a). In the interior of the Atlantic, $\Delta^{14}C_{DIC|CC}$ ranges between

-70 and -170 ‰ and decreases from the surface to the bottom, with small vertical gradients in the high latitudes (Fig. 8a). This is superimposed by a southward gradient of $\Delta^{14}C_{DIC}$. The meridional $\Delta^{14}C_{DIC}$ gradient reverses in the Pacific. Different to the northern North Atlantic, there is no evidence of deep sea ventilation in the North Pacific where our model arrives at minimum $\Delta^{14}C_{DIC|CC}$ values exceeding -290 ‰ (Fig. 8a).

Overall, simulation CC captures the large-scale distribution of pre-nuclear $\Delta^{14}C_{DIC}$ as reconstructed by GLODAPv1.1 (Fig. 7a-b, 8a-b). However, at 50 m depth the simulated $\Delta^{14}C_{DIC}$ is too high by 10 – 30 ‰ in the low- and mid-latitudes, and too low by about the same amount in the high latitudes (see Fig. 9a; according to GLODAP $\Delta^{14}C_{DIC}$ ranges from -50 to -170 ‰ with -71 ‰ on average). In the interior of the oceans, our model results are typically 20 – 60 ‰ too low (Fig. 10). The excessive depletion reaches



-70 ‰ in the abyssal North Atlantic and in the North Pacific at 3 km depth (Fig. 10a). In the upper layers of the oceans, the GLODAPv1.1 data probably reflect the $^{14}$C Suess effect (Suess, 1955), which is not considered in our simulations. The excessive depletion in the deep sea indicates that the simulated AMOC leads to overly shallow and weak ocean ventilation, which is enhanced by the progressive radioactive decay of DI$^{14}$C along its passage through the interior of the ocean.

Different to $\delta^{14}$C$_{DIC}$, $\Delta^{14}$C$_{DIC}$ is corrected for isotopic fractionation. In practice (as well as in the comprehensive DI$^{14}$C cycle modelling approach CC), the correction is made after the simultaneous determination of $\delta^{13}$C$_{DIC}$ and $\delta^{14}$C$_{DIC}$. In the inorganic $^{14}$C (IC) modelling approach the fractionation of $^{14}$C is omitted beforehand, so that posterior corrections are not necessary. That is, $\delta^{14}$C$_{DIC|IC}$ equals $\Delta^{14}$C$_{DIC|IC}$ which should equal $\Delta^{14}$C$_{DIC|CC}$. As the IC approach also neglects the radioactive decay of

organic carbon, it considers seven tracers less than the CC approach which is accompanied by an increase in model speed (simulated years per day) of about 15 %.

At 50 m depth $\Delta^{14}$C$_{DIC|IC}$ ranges from -50 ‰ to -160 ‰ with an average value of 72 ‰ (Fig. 7c). Similar to experiment CC, the highest values of $\Delta^{14}$C$_{DIC|IC}$ are found in subtropical surface waters. The lowest surface water values of $\Delta^{14}$C$_{DIC|IC}$ are also found in the Southern Ocean. In the interior of the oceans, the

isotopic depletion with respect to the atmosphere ranges from -90 ‰ in the North Atlantic to -310 ‰ in the North Pacific (Fig 8c). Comparing IC with the GLODAPv1.1 reconstruction, we find that the enrichment of subtropical surface values is less pronounced in IC while the depletion in the high-latitudes increases (Fig. 9b). The latter is also the case in the interior of the oceans (Fig. 10b). Most notably, the outcomes of experiment IC are everywhere lower (by up to 30 ‰) than the results of simulation CC (Figs.

11a, 11c) which is explained as follows.

Analogously to sensitivity experiment NP, the IC approach disregards photosynthetic fractionation which leads to lower DI$^{14}$C concentrations in the euphotic zone than in simulation CC. In addition, the IC approach disregards the DI$^{14}$C enrichment of the mixed layer associated with air-sea exchange. Furthermore, since the IC approach disregards the radioactive decay of phytoplankton, the loss of DI$^{14}$C

due to photosynthesis is overestimated in the mixed layer. Therefore, preformed DI$^{14}$C is systematically lower than in simulation CC and becomes further depleted through radioactive decay in the deep sea. This bias is similar to the lower values of "abiotic" $\Delta^{14}$C compared to "biotic" $\Delta^{14}$C simulated by Frischknecht et al. (2022).

Instead of computing absolute concentrations of DIC, DI$^{13}$C, and DI$^{14}$C and converting them to $\Delta^{14}$C$_{DIC}$

a posteriori, the $\Delta^{14}$C approximation (DA) simulates $^{14}$R$_{DIC|DA}$ = 1 + 0.001 $\Delta^{14}$C$_{DIC|DA}$ as a single radioconservative tracer which is connected with the carbon cycle through the $^{14}$CO$_2$ air-sea exchange. This approach is about five times faster than the approaches CC and IC. First results of the implementation of $^{14}$R$_{DIC|DA}$ into FESOM2 were shown by Lohmann et al. (2020) for the default FESOM mesh with 127000 horizontal surface nodes. Here, we repeated the experiment, now using the low-resolution mesh

of experiments CC and IC to be able to compare the results of all approaches directly. For the same reason,





we discuss model results after 5000 simulated years but point out that experiment DA has been run over 17000 years in total. The maximum drift of $\Delta^{14}C_{DIC|DA}$ between 5000 and 17000 simulated years is about -3.5 ‰ in North Pacific Deep Water, which is much smaller than the $\Delta^{14}C_{DIC}$ differences between the various modelling approaches in this study after 5000 years shown below.

At 50 m depth $\Delta^{14}C_{DIC|DA}$ ranges from -40 to -130 ‰, with $\Delta^{14}C_{DIC|DA}$ = -58 ‰ on average (Fig. 7d). In intermediate and deep water $\Delta^{14}C_{DIC|DA}$ declines from -60 ‰ in the North Atlantic to -280 ‰ in the North Pacific (Fig. 8d). At upper levels, $\Delta^{14}C_{DIC|DA}$ is almost everywhere higher than $\Delta^{14}C_{DIC}$ according to GLODAPv1.1 (Figs. 9c, 10c). In the deep sea, $\Delta^{14}C_{DIC|DA}$ is still too low but the depletion is less pronounced than in simulations CC and IC (Fig. 10c).

Experiment DA yields the highest $\Delta^{14}C_{DIC}$ values of all the three modelling approaches (Fig. 11). The DA approach assumes that DIC concentrations are constant and homogeneous (for a rigorous treatise see Mouchet, 2013). Following Toggweiler et al. (1989), our calculation of $^{14}CO_2$ air-sea exchange fluxes in simulation DA assumes a DIC concentration of 2000 mmol m$^{-3}$ in the mixed layer which is somewhat lower than observed and simulated in most areas, most notably in the high latitudes (Fig. A5a-b). This

leads to faster and hence higher $^{14}C$ uptake in DA than in CC and IC because the $^{14}CO_2$ invasion flux is inversely proportional to the DIC concentration in the mixed layer. The absolute $\Delta^{14}C_{DIC}$ differences between DA and CC are largely less than 10 ‰ (Figs. 11b, 11d). Moreover, the relative uncertainty of the DA approach with respect to the correct DI$^{14}$C implementation is less than 5 % (Fig. 12) and actually smaller than the error of 10 % originally estimated by Fiadeiro (1982).

## 325 **4 Summary**

We have added the carbon isotopes $^{13}$C and $^{14}$C to the marine biogeochemistry model REcoM3 and tested the implementation in long-term equilibrium simulations in which the configuration REcoM3p was coupled with the ocean general circulation model FESOM2.1. Regarding the carbon-isotopic composition of DIC ($\delta^{13}C_{DIC}$ and $\Delta^{14}C_{DIC}$), our model results are largely consistent with marine $\delta^{13}C_{DIC}$ and $\Delta^{14}C_{DIC}$

fields reconstructed for the pre-anthropogenic period. The simulations also exhibit discrepancies, such as overly depleted $\delta^{13}C_{DIC}$ values in upwelling regions and excessive depletion of $\Delta^{14}C_{DIC}$ in the interior of the North Pacific. To some extent, the inaccuracies of $\delta^{13}C_{DIC}$ indicate shortcomings in modelled organic carbon cycling. The radiocarbon results ($\Delta^{14}C_{DIC}$) reflect the rather shallow overturning circulation provided by our low-resolution ocean general circulation model test configuration with idealized repeat

year climate forcing. As future simulations with a scientific focus will be carried out with considerably higher horizontal resolution and more realistic climate forcing, we expect some of the biases discussed in this study to decrease. These simulations should also consider transient boundary conditions of $^{13}$C and $^{14}$C which provide additional benchmarks for the model. For these reasons we did not attempt to further



tune REcoM3p here, e.g. by adjusting semi-empirical biogeochemical parameters such as gas transfer
velocity or biogenic fractionation coefficients.

As $\Delta^{14}C_{DIC}$ is dominated by radioactive decay and transport processes, we have additionally explored the accuracy of two simplified modelling approaches which are more efficient than the complete consideration (CC) of the DI$^{14}$C cycle. One approach (IC) neglects isotopic fractionation but still considers biological processes. Another approach (DA) only considers the DI$^{14}$C / DIC ratio for constant
and homogeneous DIC concentrations and further disregards the marine carbon cycle. The relative uncertainty between the comprehensive and simplified approaches is less than 5 %. Therefore, the simplified $\Delta^{14}C_{DIC}$ modelling approaches should be sufficiently accurate for radiocarbon dating of marine climate archives.

*Code availability.* The source code is available at https://doi.org/10.5281/zenodo.8169243.

*Author contributions*. MB developed the isotope code, conducted the simulations and prepared the manuscript with contributions from all co-authors.

*Competing interest.* The authors declare that they have no conflict of interest.

*Acknowledgements.* This work was supported by the German Federal Ministry of Education and Research (BMBF) through the PalMod project (grant number: 01LP1919A) which is part of the Research for Sustainability initiative FONA (https://www.fona.de). MB is additionally funded through DFG-ANR
project MARCARA. JH and OG were funded by the Initiative and Networking Fund of the Helmholtz Association (Helmholtz Young Investigator Group Marine Carbon and Ecosystem Feedbacks in the Earth System [MarESys], grant number VH-NG-1301). We thank D. Sidorenko for FESOM model support.



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





# Figures

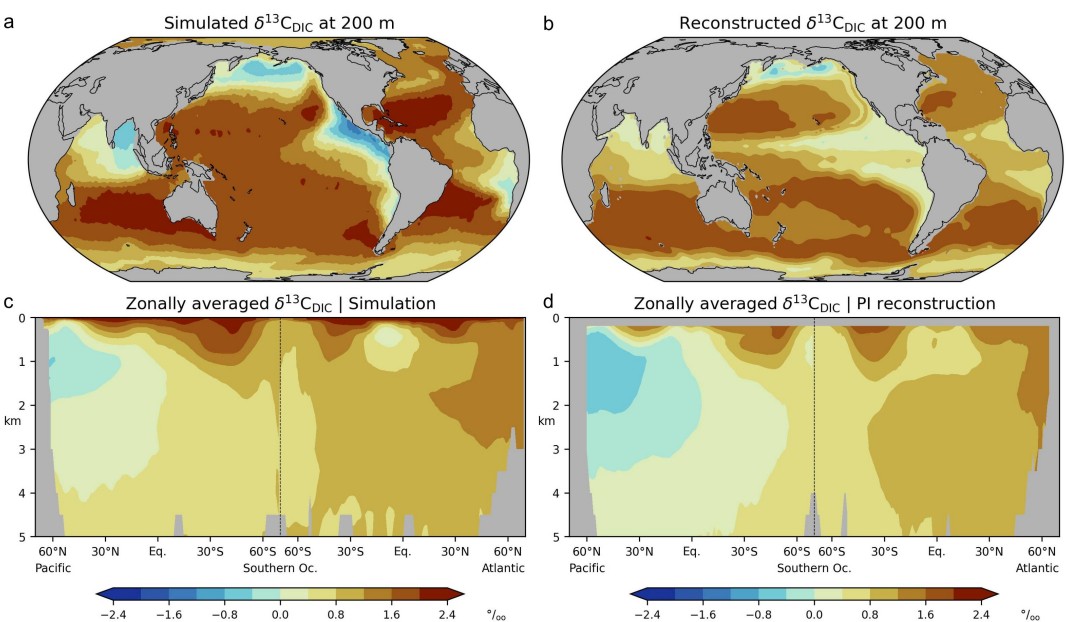

**Figure 1.** Preindustrial δ¹³C of DIC, (a, c): this study, (b, d): reconstruction (Eide et al., 2017). (a, b): values at 200 m depth, (c, d) zonal-mean values in the Atlantic and Pacific. Map projection here and in other figures is area-preserving (Equal Earth projection, Šavrič et al., 2019)



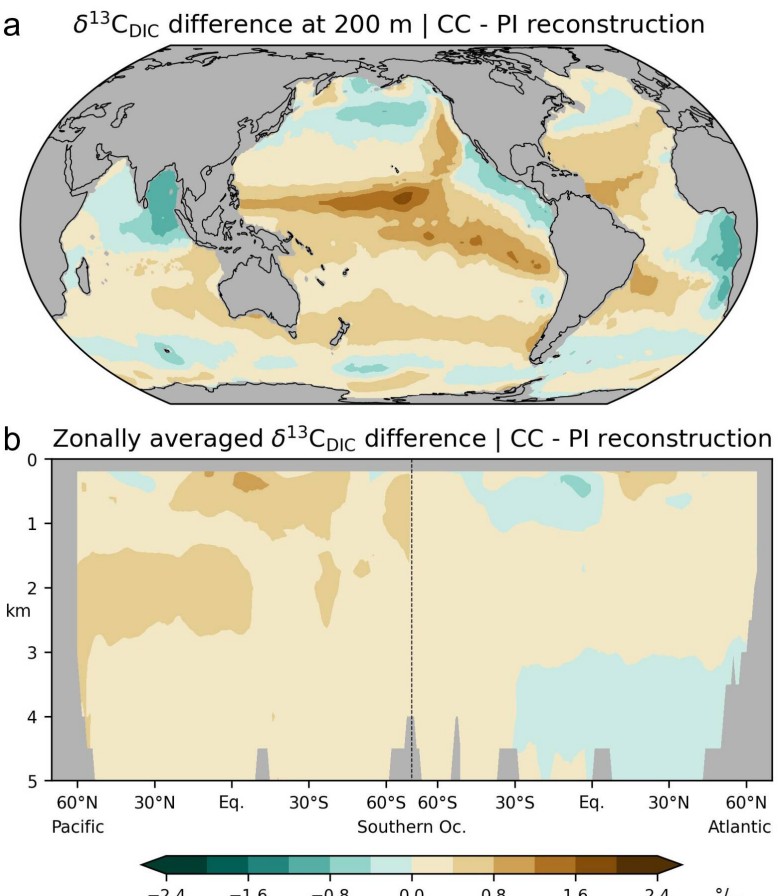

**Figure 2.** Differences between simulated (this study; CC) and reconstructed (Eide et al., 2017; PI) $\delta^{13}C$ of DIC for the preindustrial period, (a): at 200 m depth, (b): zonal-mean values in the Atlantic and Pacific.





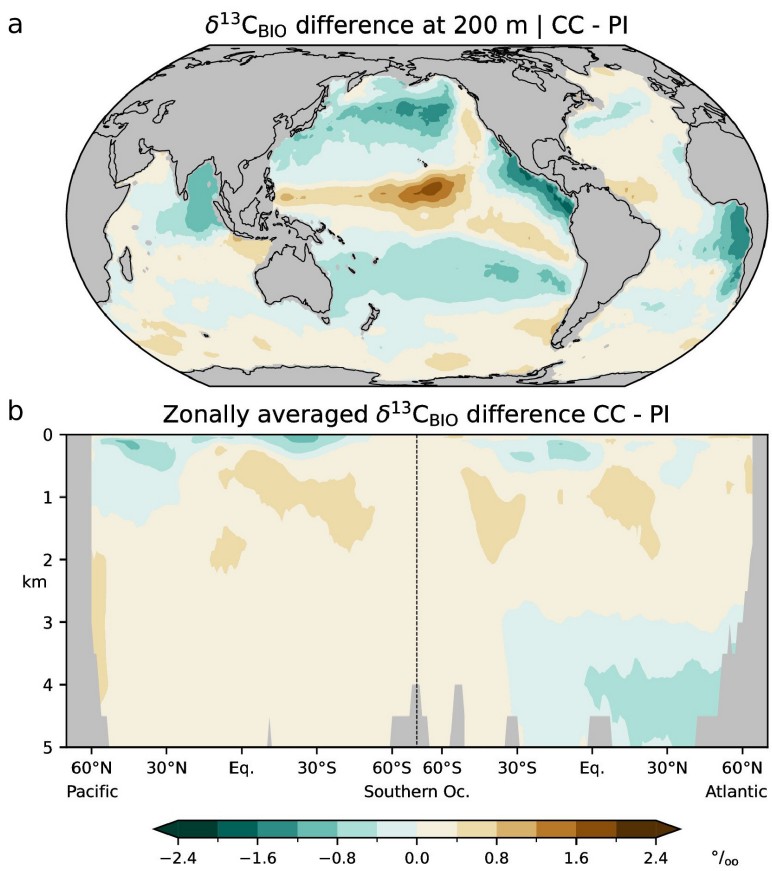

**Figure 3.** Differences between simulated (this study; CC) and estimated (Eide et al., 2017; PI) $\delta^{13}C_{BIO}$ (the biological component of $\delta^{13}C_{DIC}$ in the absence of air-sea exchange) during the preindustrial period, (a): at 200 m depth, (b): zonal-mean values in the Atlantic and Pacific.






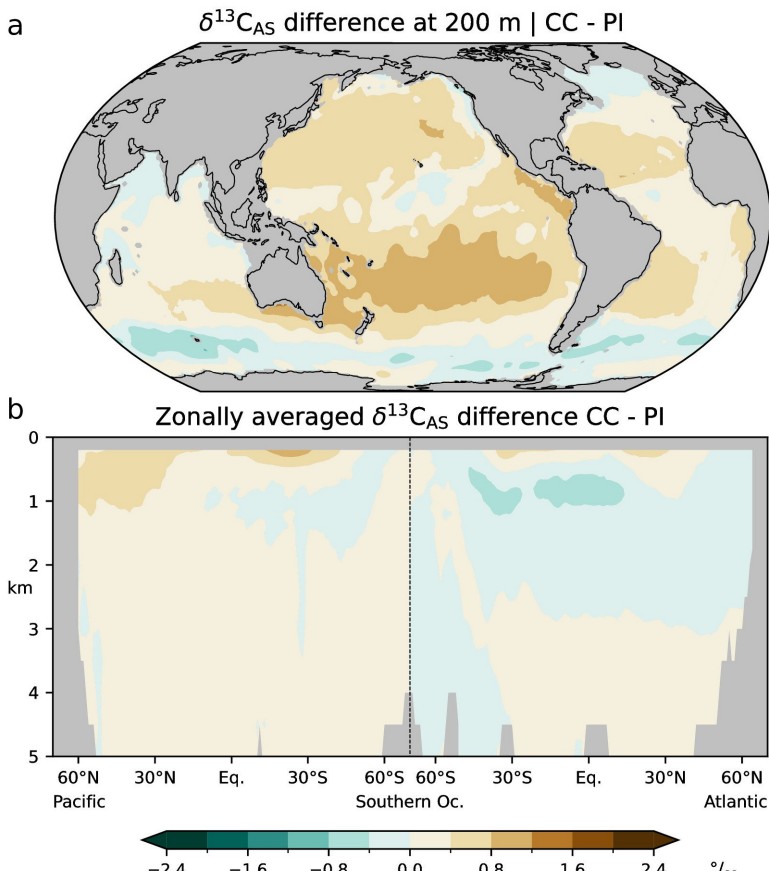

**Figure 4.** Differences between simulated (this study; CC) and estimated (Eide et al., 2017; PI) $\delta^{13}C_{AS}$ = $\delta^{13}C_{DIC}$ - $\delta^{13}C_{BIO}$ during the preindustrial period, (a): at 200 m depth, (b): zonal-mean values in the Atlantic and Pacific.



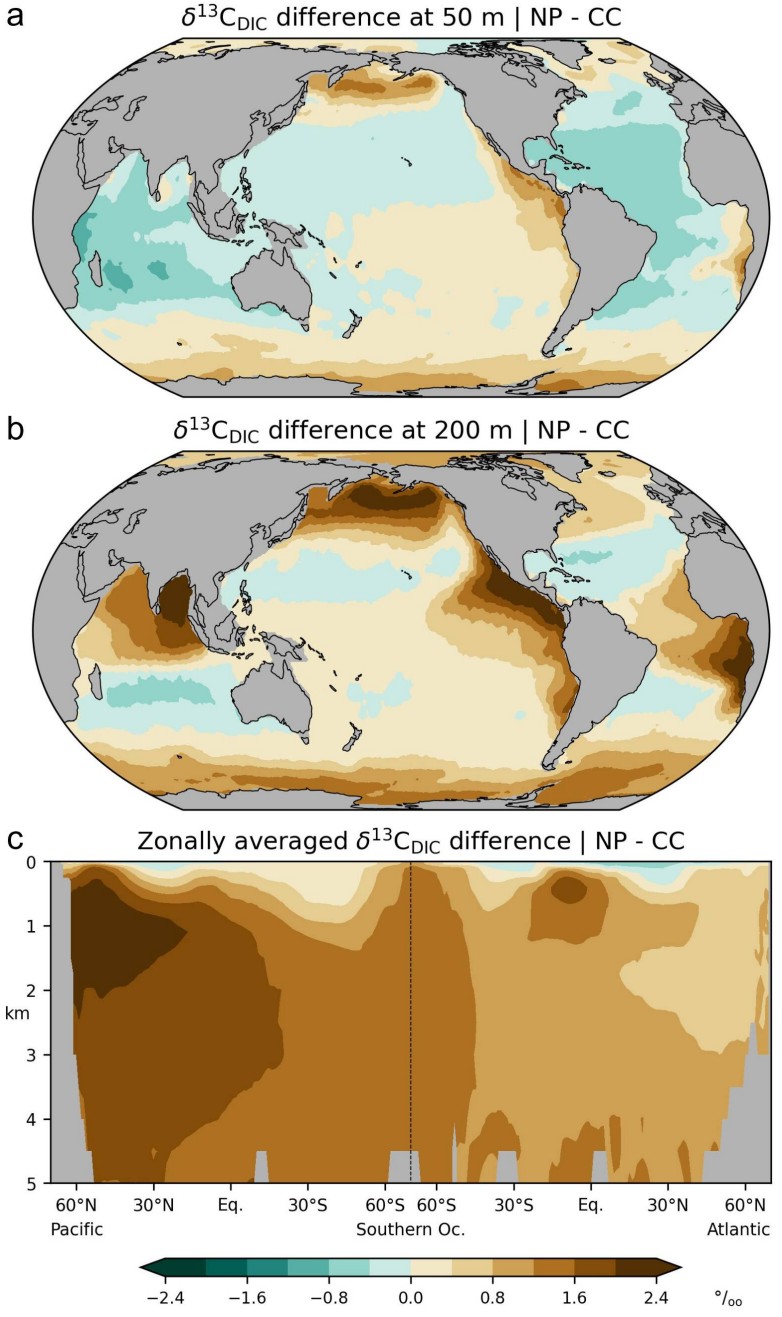

**Figure 5.** Changes in preindustrial δ¹³C of DIC if isotopic fractionation during photosynthesis is disabled (sensitivity experiment NP versus simulation CC), (a): at 50 m depth, (b) at 200 m depth, (c): zonal-mean values in the Atlantic and Pacific.



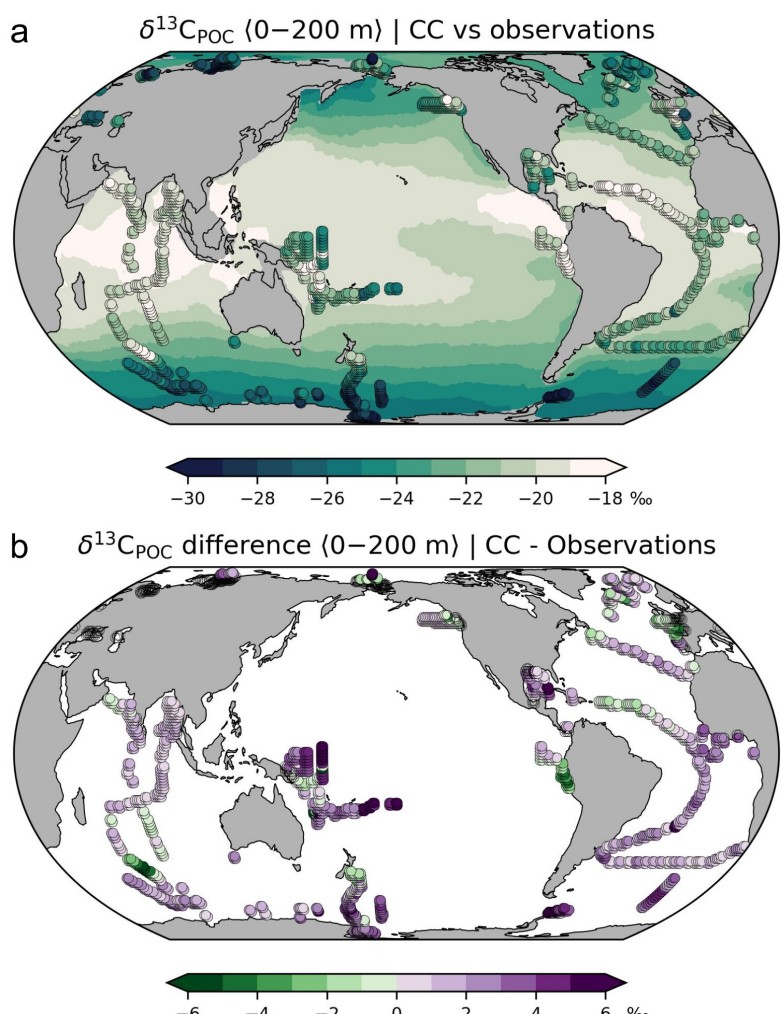

**Figure 6.** (a) $\delta^{13}$C of POC averaged over the upper 200 m. Shaded areas: simulation results (this study),
dots: bulk matter observations for the period 1964-2015 (compilation by Verwega et al., 2021; see further
references therein). (b) Differences between simulated and observed $\delta^{13}$C of POC.





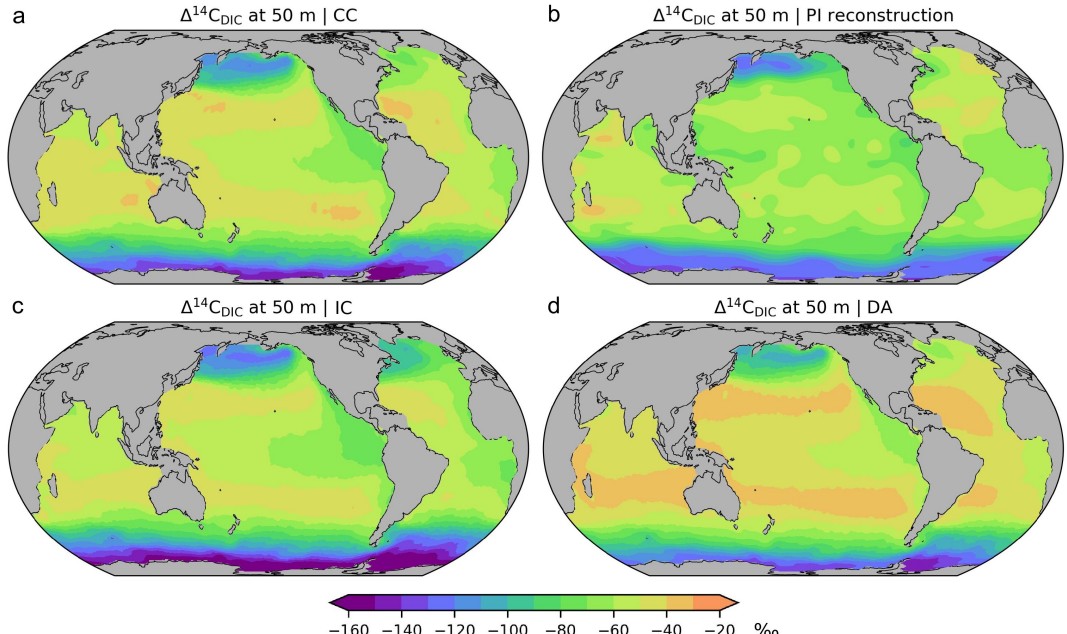

**Figure 7.** Preindustrial $\Delta^{14}$C of DIC at 50 m depth, (a): simulation CC considering the complete marine $^{14}$C cycle, (b): reconstruction (Key et al., 2004), (c): simulation IC applying the inorganic $^{14}$C approximation, (d): simulation DA applying the $\Delta^{14}$C approach. See the main text for further explanations.





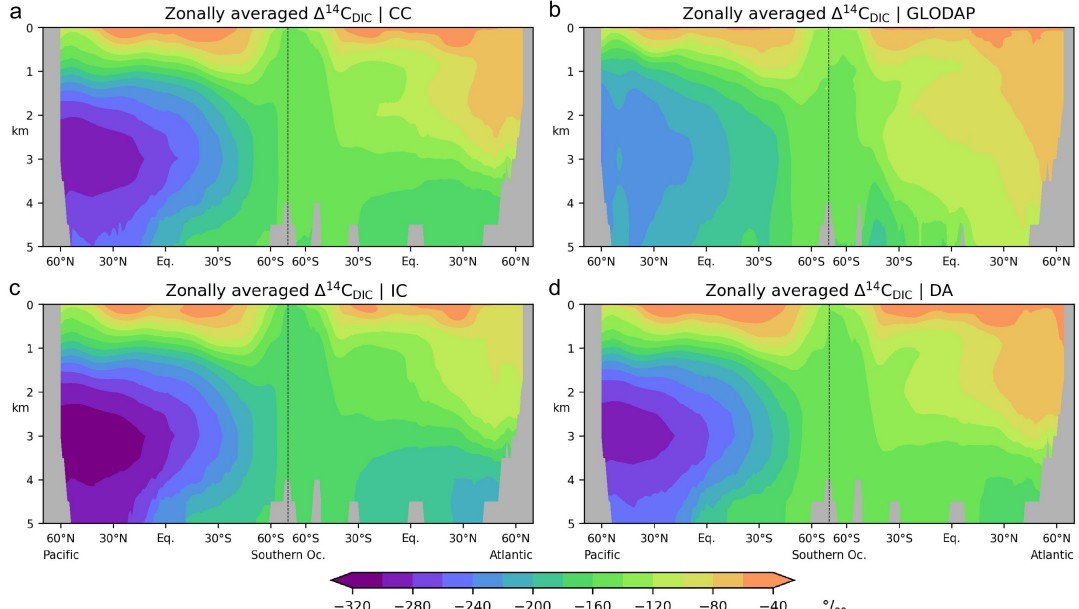


**Figure 8.** Preindustrial $\Delta^{14}$C of DIC in the Atlantic and Pacific. (a): Simulation CC, (b): reconstruction (Key et al., 2004), (c): simulation IC, (d): simulation DA. See the main text for further simulation explanations. Note the different colour scale ranges in Figures 7 and 8.



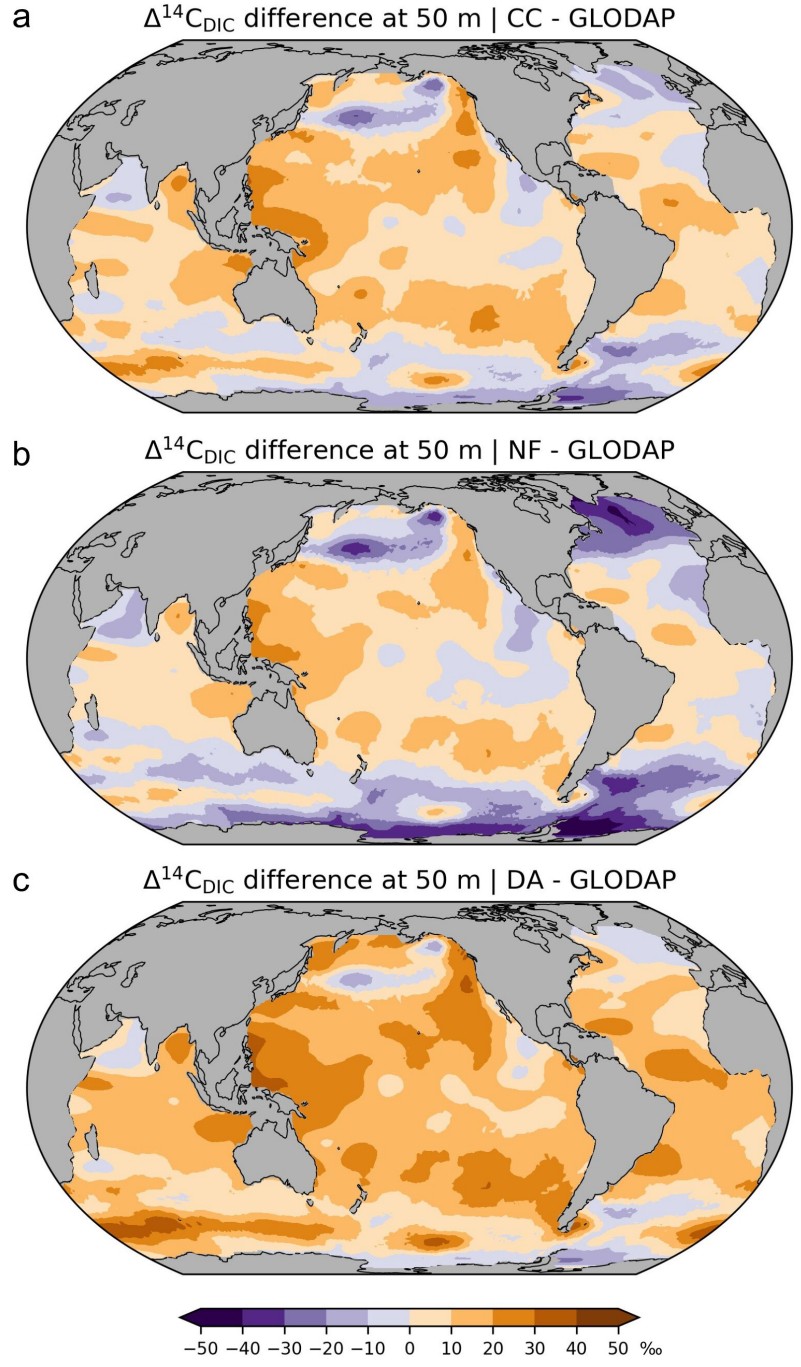


**Figure 9.** Differences between simulated and reconstructed preindustrial $\Delta^{14}$C of DIC at 50 m depth, (a): simulation CC minus reconstruction (GLODAP; Key et al., 2004), (b): simulation IC minus reconstruction, (c): simulation DA minus reconstruction.





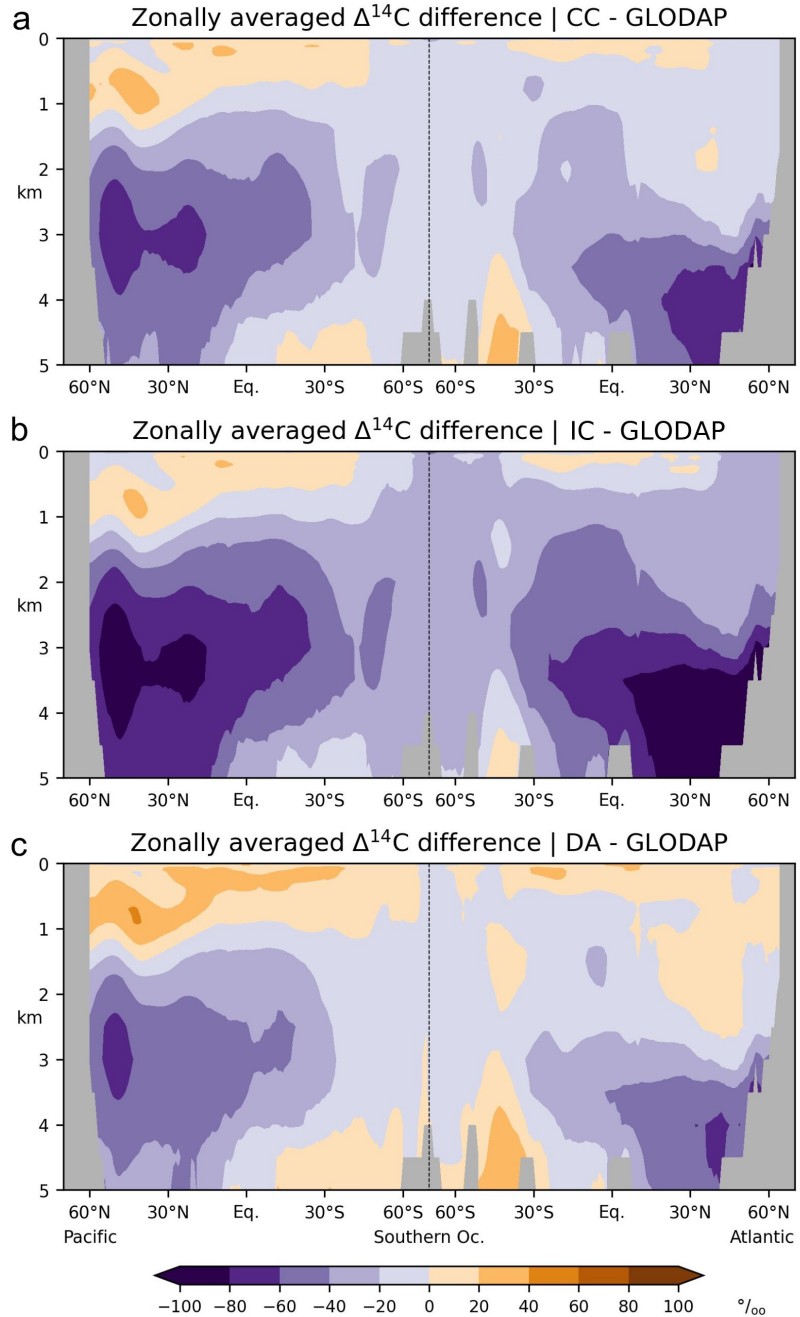


**Figure 10.** Differences between simulated and reconstructed preindustrial $\Delta^{14}C$ of DIC in the Atlantic and Pacific, shown are zonal-mean values, (a): simulation CC minus reconstruction (GLODAP; Key et al., 2004), (b): simulation IC minus reconstruction, (c): simulation DA minus reconstruction. Note the different colour scale ranges in Figures 9 and 10.




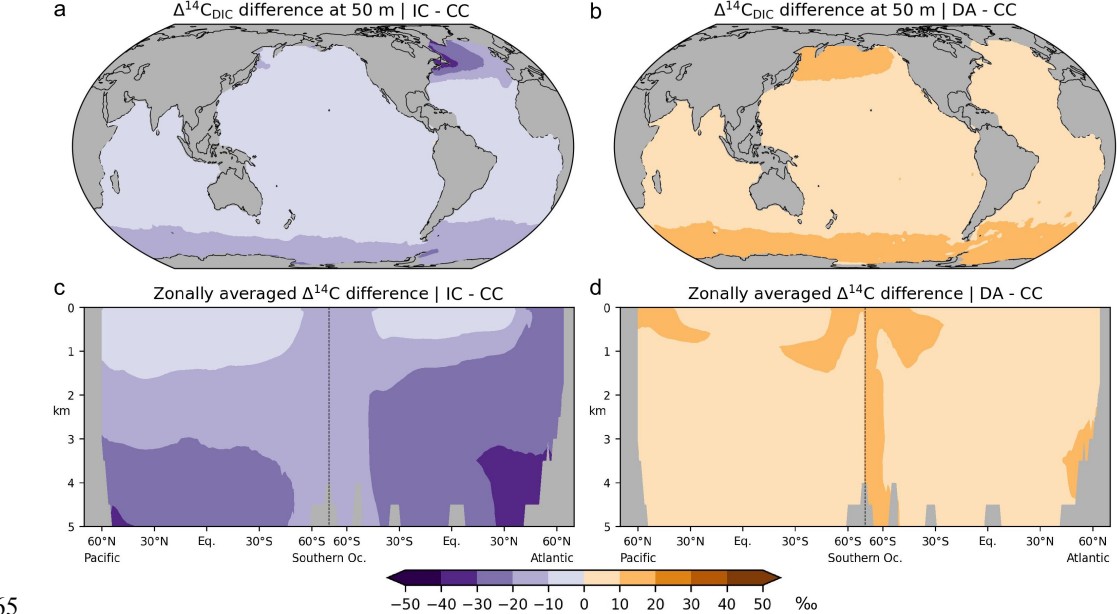

**Figure 11.** Absolute differences in preindustrial $\Delta^{14}C_{DIC}$ between the various simulation approaches, shown are results at 50 m depth (a, b) and in the Atlantic and Pacific (c, d). (a, c): Inorganic $^{14}C$ (IC) modelling approach versus complete $^{14}C$ cycle (CC), (b, d): $\Delta^{14}C$ approximation (DA) versus simulation

CC.





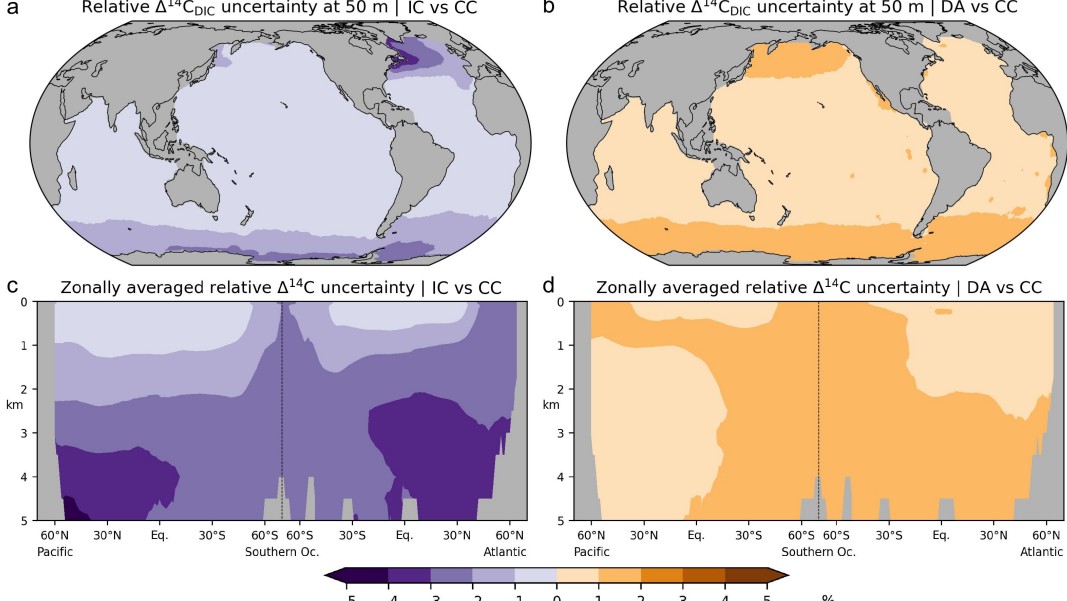

**Figure 12.** Relative differences in preindustrial $\Delta^{14}C_{DIC}$ between the various simulation approaches, shown are absolute values at 50 m depth (a, b) and in the Atlantic and Pacific (c, d). (a, c): No-fractionation (IC) approach versus complete $^{14}C$ cycle (CC), (b, d): $\Delta^{14}C$ approximation (DA) versus simulation CC.



# Table

**Table 1.** List of model experiments discussed in this paper.

| Name | Description | $^{13}$C | $^{14}$C | Notes |
|------|-------------|----------|----------|-------|
| CC | Complete $^{14}$C cycle | Yes | DI$^{14}$C | Control experiment |
| | | | DO$^{14}$C | |
| | | | PI$^{14}$C | |
| | | | PO$^{14}$C | |
| IC | Inorganic $^{14}$C only | Yes | DI$^{14}$C | Isotopic fractionation of $^{14}$C is also neglected |
| NP | No isotopic fractionation during photosynthesis | Yes | As in IC | Sensitivity experiment to study $\delta^{13}C_{DIC}$ |
| DA | $\Delta^{14}$C approximation | No | $\Delta^{14}C_{DIC}$ | Without REcoM3, only FESOM2.1 |



 **Appendix A**

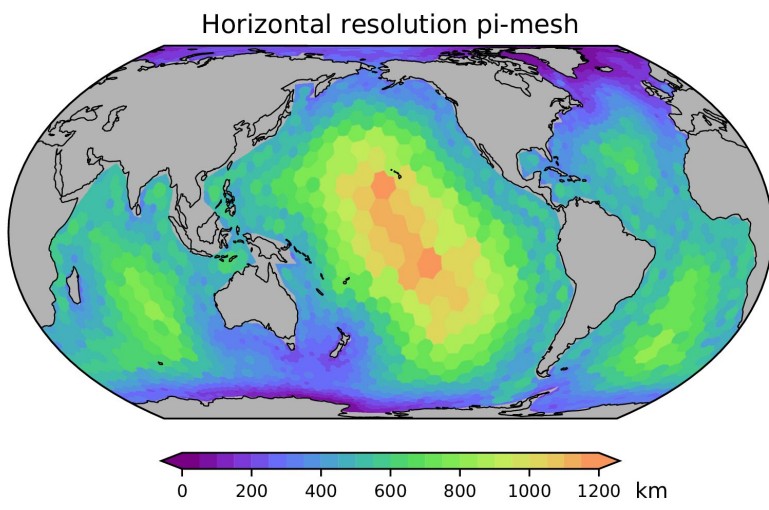

**Figure A1.** Horizontal resolution of FESOM2.1-REcoM used in this study. The mesh has 3140 surface

nodes. See also https://fesom.de/models/meshessetups/ for an impression of the bathymetry.




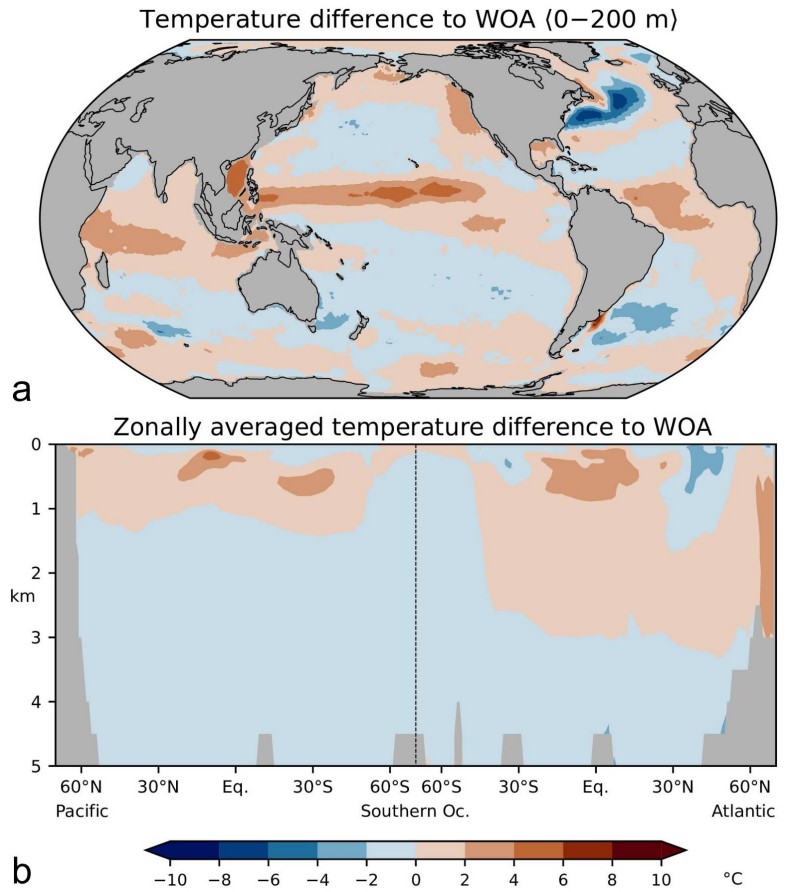

**Figure A2.** Differences between simulated and observed (Locarnini et al., 2010) temperatures, (a) averaged over the upper 200 m, (b) in the Atlantic and Pacific. See Scholz et al. (2019) for comparison with higher-resolution simulations.





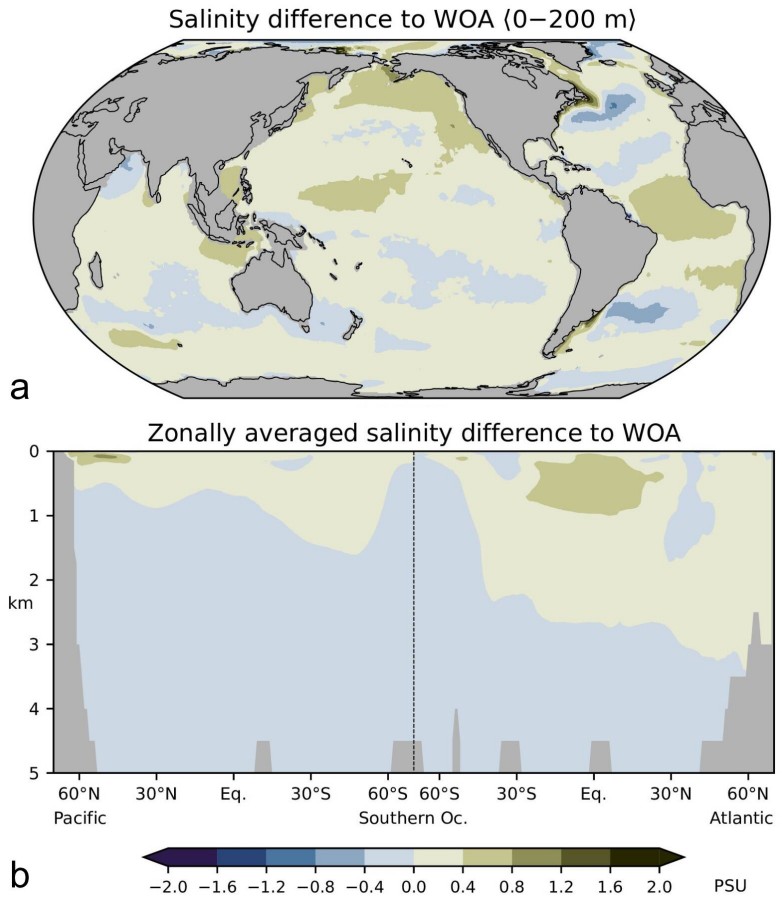

**Figure A3.** Differences between simulated and observed (Antonov et al., 2010) salinities, (a) averaged
over the upper 200 m, (b) in the Atlantic and Pacific. See Scholz et al. (2019) for comparison with higher-
resolution simulations.




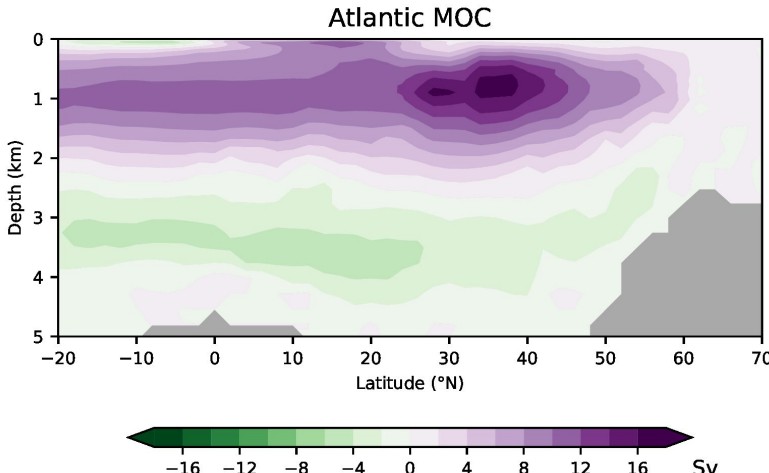

**Figure A4.** Simulated meridional overturning circulation (MOC, 1 Sv = $1 \times 10^6$ m$^3$ s$^{-1}$) in the Atlantic.
See also Scholz et al. (2019, 2022) for comparison with higher-resolution simulations.

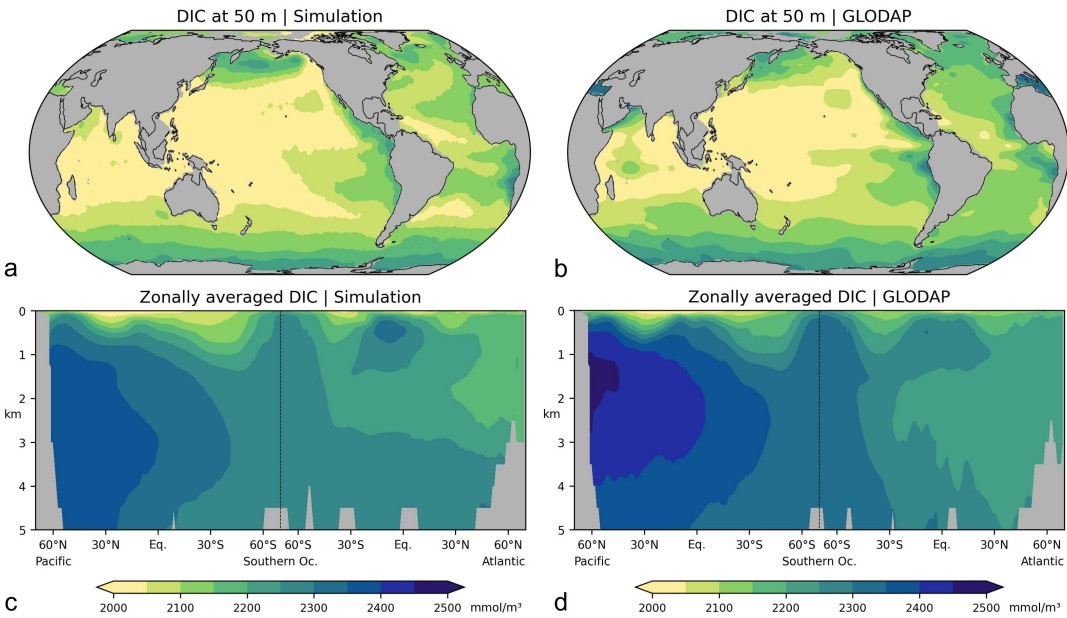

**Figure A5.** Concentrations of dissolved inorganic carbon (DIC), (a, c): this study, (b, d): observations for
1972 - 2013 CE, normalized to the year 2002 (Key et al., 2015; Lauvset et al., 2016). (a, b): Concentrations
at 50 m depth, (c, d) zonal-mean values in the Atlantic and Pacific. Model results are interpolated to the
resolution of the observations (1° x 1° x 33 layers).