# Peer review of "Carbon isotopes in the marine biogeochemistry model FESOM2.1-REcoM3"

_EGUsphere, 2023_

## Referee Comment (RC1)

**Review 'Carbon isotopes in the marine biogeochemistry model FESOM2.1-REcoM3' by Butzin et al. for Geoscientific Model Development**

**General comments**

Dear Editor, dear Butzin et al.,

This manuscript on the implementation of the [13]C and [14]C isotopes of carbon in the FESOM2.1-REcoM3 model is well-written, concise and describes a timely development of this model as other Earth System Models have done the same in recent years. Implementation of these C isotopes in a model of this complexity allows for exciting new studies in both paleoceanography and contemporary global carbon cycling.

The control model setup ('called 'CC') is generally described in enough detail to ensure reproducibility. The authors have provided many figures to document their results. The authors compared to observational datasets and show that both radiocarbon and $\delta^{13}$C show too large gradients (vertically as well as along the pathway of overturning circulation) in the model. They also discuss drivers of these biases. Besides their control experiment 'CC', Butzin et al. also explored effects of some other model setups, such as the absence of fractionation during photosynthesis ('NP') and more efficient versions of the radiocarbon code ('IC' and 'DA'). They thereby introduce new modelling approaches of radiocarbon, addressing one of the major issues with ([14])C isotope modelling – the computational cost.

The most important points I would like to raise are as follows.

1. The definitions of the $\delta^{13}$C/$\delta^{14}$C raise some questions such as which standards are followed (PDB/VPDB), and which constants are used (following OMIP or not?). See detailed comments on L44-50.
2. A direct comparison is made with Eide et al. (2017) their PI $\delta^{13}$C dataset. This dataset is the result of a subtraction of an estimate of the Suess effect from observational data. It has been shown that the Suess effect is likely underestimated by Eide et al. (2017) in Liu et al. (2021). I think it is important that in the comparison this underestimation is considered and discussed.
3. The $\delta^{13}C_{BIO}$ calculation is based on observational data, whereas a model-based separation of $\delta^{13}C_{BIO}$ and $\delta^{13}C_{AS}$ would be internally consistent and just as easy to calculate. See detailed comments on L222.

I would recommend publication of this manuscript after addressing these and the following minor points.

Yours truly,

Anne Morée

**Comments**

**Introduction**

L 37-38: 'numerous ocean general circulation models have been equipped with carbon isotopes and applied in Earth system modelling studies' marine biogeochemistry models have been equipped with C isotopes, right? And only part of your references is for applications in actual ESMs? I think this is an opportunity to highlight it is still quite unique to have C isotopes in an ESM.

L42: $^{12}$C is also a C isotope, so the use of the word 'both' is inaccurate here.

L44-50:
- I think it is relevant to state which standard you use (Pee Dee Belemnite for $\delta^{13}$C), as the new standard is actually the VPDB (Vienna Pee Dee Belemnite) although this is usually not implemented by ESMs because many paleorecords are still reported in the PDB standard.
- For $\delta^{14}$C, the CMIP6 standard is 1.170e$^{-12}$ and Karlén et al. (1964) is outdated (Orr et al., 2017, see page 2194). Please check in general whether you have followed CMIP6 guidelines, and state in the article if you have deviated from it (this is already done several places but may need to be extended).
- Add a promille symbol after all '1000' in Eq1-3.
- How is $^{12}$C calculated (total modelled C minus $^{13}$C?)?

**Model description**

L63: a sediment model is included, are the C isotopes also in there? If so, how were they initialized? And how is the drift at the end of the control simulation?

L77-79: Are they passive tracers? If $^{13}$C and $^{14}$C are included in sinking of organic matter, they are not passive in my point of view. Could you specify all carbon compounds which you have included the C isotopes in (e.g., POC, DOC, DIC, PIC, CaCO3, phytoplankton C, zooplankton C?)

L111: 0.014 should be 0.0144 (Orr et al., 2017)

L119-121: You could also refer here to e.g., Liu et al. (2021) and some other studied as they have explored the difference between these different formulations in MPI-ESM.

L124: what makes it 'robust'?

L138: In Craig et al. (1954, page 133; https://www.jstor.org/stable/3af8a654-6d9e-38ec-9358-ba8b25f2a7c1?seq=19) it states 'The fractionation factors for $^{14}$C will then be the square of the $^{13}$C factors, and, since these numbers are close to 1, the enrichment (fractionation-1) of $^{14}$C in a given compound should be almost exactly twice that of $^{13}$C in both equilibrium and rate reaction isotopic effects.' Why use the approximation here instead of the square (i.e., $\alpha^{14}$C= $(\alpha^{13}$C$)^2$ )? This power of 2 is actually uncertain as well (see detailed discussion on the value on this 'fractionation ratio' in Fahrni et al., 2017; https://www.sciencedirect.com/science/article/pii/S0016703717303344?via%3Dihub).

L138-142: Could you elaborate here what the advantages are of and the reasons for specifically doing this set of experiments? The details of the experiments are mostly given in Sections 3.2 and 3.3, maybe bring them up to here? Or bring forward L341-345?

L169-170: Which overturning circulation metrics did you look at for the drift: AMOC? Pacific overturning/Drake Passage? What is the remaining drift in both biogeochemistry (particularly also the C isotopes) and physical state after the full 6000 years of simulation?

L175-177: If forced with atmospheric concentrations, how do you ensure mass balance?

L181-186: Could you report approximate bias magnitude here for these water masses as well and reflect on how such biases compare to other models?

L190-193: How does the biogeochemical state otherwise compare to observations? E.g., Apparent Oxygen Utilization or phosphate, which correlate strongly with $\delta^{13}C$?

**Carbon-13**
L195-196: 'meridional sections': The figures do not show sections but zonal means, please provide the longitude range information (or basin mask?) you have used to make these plots and clarify here.

L199: 'in wide areas', do you mean over a large part of the ocean at 200m depth?

L206-213: The patterns in Fig. 1 look good; if you would subtract the global mean bias from your model (which you could argue for, especially if you have remaining drift), how good is your agreement then?

L214: In other studies, $\Delta\delta^{13}C_{DIC}$ is used to designate the vertical marine $\delta^{13}C$ gradient (random relatively recent example: https://www.nature.com/articles/s41561-019-0473-9). I think the use of '$\delta^{13}C$ bias' or something similar would prevent confusion.

L222: Equation 10, which you have taken from Eide et al. (2017) and adjusted to be able to use DIN, has constants based on observational data as described in Eide et al. (2017). When using a model however, you should in my opinion use the full equation by Maier-Reimer et al. (1992) (see also equation 3 in Eide et al. (2017)), in which you can then insert the model specific parameters. These parameters likely deviate quite a bit from your observational-based Equation 10 (see e.g., Morée et al., 2018; https://bg.copernicus.org/articles/15/7205/2018/bg-15-7205-2018.html), text after Eq. 3). When updating this, the comments made in Lines 231-234 should be updated as well.

L236: 'biogenic fractionation', you define $_{13}\alpha_p$ before as '$_{13}\alpha_p$ is the isotopic fractionation factor associated with photosynthesis', please be consistent.

L226-230: Could you quantify here (e.g., globally or by region if preferable) what percentage of the bias in $\delta^{13}C_{DIC}$ is due to the $\delta^{13}C_{BIO}$ bias, and what from the residual $\delta^{13}C_{AS}$? This would really highlight where further attention is most needed to reduce mean bias. You could then also add this to the summary at L332.

**Radiocarbon**

L261-262: 'This is superimposed by a southward gradient of $\Delta^{14}C_{DIC}$. The meridional $\Delta^{14}C_{DIC}$ gradient reverses in the Pacific.' I do not really follow this. Specify direction of gradient (negative toward south in Atlantic). What reversal do you see in the Pacific?

L264: what maximum water mass radiocarbon age does that represent? Is your model relatively slowly overturning and therefore relatively old compared to observations (how much older in N-Pacific in terms of e.g. ideal age/radiocarbon age?)? You mention AMOC in L272-274, but formation rates and export of southern source waters would be relevant for maximum water mass age as well outside the north Atlantic.

L265-270: Could you also here (and possibly at several points in this section) report the bias in terms of radiocarbon age, which may be more intuitive to understand for some readers (and quite comparable to ideal age tracers which almost all models have)?

L301: What is a radioconservative tracer? You need $\delta^{13}C$ for the calculation of $\Delta^{14}C$, how do you go about that? I have not understood this experiment based on the description here.

L323: 'the correct $DI^{14}C$ implementation', do you mean your CC experiment?

**Summary**

L330-332: I think this sentence does not really summarize your biases. More than the low simulated (CC) $\delta^{13}C$ in upwelling zones, I think the $\delta^{13}C$ and radiocarbon biases are summarized by generally too steep vertical gradients (which leads to upwelling of too-depleted waters for $\delta^{13}C$) as well as too depleted waters at the 'end' of the overturning circulation (as your model overturns relatively slowly). This comment also applied to lines 14-16.

L345-346: I think the bias introduced by using the simplified approaches for modelling radiocarbon should be discussed not just relative to experiment CC but also relative to the PI data: I.e., from Fig. 10 it is visible that the already existing bias (too steep gradients) gets even stronger in the simplified approaches.

**Other points**

L619: Here you specify which model experiment you have used (CC), can you do so for all Figs. (e.g., Fig. 1)?

L 688&694&699, etc.: If the figure considers WOA data, please specify and cite which WOA data you have used. For all zonal means, please specify over which longitudes the zonal means were taken or whether e.g., some basin mask was used. Instead of showing model and observational data side-to-side, I think it easier to see the differences by showing the model-observation bias like you do in Figs. 2 and 3 (and if you wish to also show the absolute values, keep the model plots too).

---

## Author Comment (AC1)

*Reviewer text is in black, author replies are in blue italics.*

This manuscript describes results from a new implementation of carbon isotopes in an ocean model. The paper is well written, nicely illustrated and the conclusions are backed up with the evidence provided. I only have a few minor comments and leave it to the discretion of the authors how much they want to change the manuscript.

*We thank Andreas Schmittner for his constructive and friendly review and are happy to answer his comments. Corresponding manuscript changes are highlighted in blue in the revision.*

Line 197: The reconstruction by Kwon et al. (2022, https://doi.org/10.1038/s43247-022-00388-8) includes the surface and could be used to compare with the model results there.

*Unfortunately, Kwon et al. (2022) do not provide gridded data and their reconstruction has spatial gaps. We think that the horizontal resolution of our model setup is too coarse for a robust comparison with local observations / reconstructions. Therefore, we stick to the rather smooth, remapped reconstruction by Eide et al. (2017). For the sake of completeness, we show preindustrial $\delta^{13}C_{DIC}$ simulated for surface water with the corresponding values by Kwon et al. (2022; dots). This figure will also be included into the Appendix of the revision (new Fig. A6) and briefly discussed in L215-217.*

[Figure]

**New Figure A6.** *Preindustrial $\delta^{13}C_{DIC}$ of surface water at about 18 m depth. Shaded areas: Simulation CC, filled circles: Reconstructed values by Kwon et al. (2022).*

*The simulation results are largely in line with the reconstructed values by Kwon et al. The model results appear to be lower than the Kwon data in the South Pacific. However, Kwon's reconstruction also exhibits higher $\delta^{13}C_{DIC}$ values than Eide et al. in the Southern hemisphere thermocline and intermediate which can be seen the figure below and which is also discussed by Kwon et al. (2022):*

[Figure]

*Preindustrial $\delta^{13}C_{DIC}$ of seawater at 250 m depth. Shaded areas: Reconstruction by Eide et al. (2017), filled circles: Reconstructed values by Kwon et al. (2022).*

Line 218: The decomposition by Broecker and Maier-Reimer (1992) is problematic as it ignores the effect of differences in preformed $\delta^{13}C$ and $PO_4$. Interior ocean $\delta^{13}C$ and $PO_4$ include preformed components. For $\delta^{13}C$ the preformed component is impacted by air-sea gas exchange, whereas for $PO_4$ it isn't. Thus $\delta^{13}C_{BIO}$ is not equal to $\delta^{13}C_{rem}$ (which doesn't include a preformed component). $\delta^{13}C_{BIO}$ doesn't include the correct biological preformed component of $\delta^{13}C$ either since it was calculated using $PO_4$. In other words, this decomposition, although widely used, is pretty much useless to understand $\delta^{13}C$.

*See our next response.*

Line 223: Since $\delta^{13}C_{BIO}$ (as calculated following Broecker and Maier-Reimer, 1992) includes effects of air-sea gas exchange (from preformed $\delta^{13}C$), $\delta^{13}C_{AS}$ includes effects of biology.

*We agree that $\delta^{13}C_{BIO}$ and $\delta^{13}C_{AS}$ should not be confused with remineralized and preformed $\delta^{13}C$ and mention this in the revision in L249-251. The dependence of $\delta^{13}C_{AS}$ from biological effects was already mentioned in the submission (now in L264-265). As mentioned in our response to Review #1, the comparison of simulated $\delta^{13}C_{BIO}$ and $\delta^{13}C_{AS}$ with the values reconstructed by Eide et al. is primarily intended as a tentative validation with further datasets in addition to $\delta^{13}C_{DIC}$ but not as a quantitative analysis of the contributions of different drivers to $\delta^{13}C_{DIC}$. This is clarified in the revision in L253-259.*

Line 247: Why not run model with Suess effect included for a proper comparison to observations? Suess effect can be expected to decrease $\delta^{13}C_{POC}$ by about 2 permil (e.g. Fig. 8 Schmittner et al., 2013). What is the global difference (model-obs)?

*The global RMS difference between modelled and observed $\delta^{13}C_{POC}$ is 2.6‰ which is mentioned in the revision at L278. The difference is higher than what could be expected from the Suess effect only. We think that a rigid analysis of this issue should deserve a dedicated separate study which, in addition to higher*

*spatial resolution, should involve more sophisticated climate forcing, anthropogenic $^{14}$C, and maybe further anthropogenic ocean ventilation tracers.*

---

## Author Comment (AC2)

*Reviewer text is in black, author replies are in blue italics.*

This manuscript on the implementation of the $^{13}$C and $^{14}$C isotopes of carbon in the FESOM2.1- REcoM3 model is well-written, concise and describes a timely development of this model as other Earth System Models have done the same in recent years. Implementation of these C isotopes in a model of this complexity allows for exciting new studies in both paleoceanography and contemporary global carbon cycling.

The control model setup ('called 'CC') is generally described in enough detail to ensure reproducibility. The authors have provided many figures to document their results. The authors compared to observational datasets and show that both radiocarbon and $δ^{13}$C show too large gradients (vertically as well as along the pathway of overturning circulation) in the model. They also discuss drivers of these biases. Besides their control experiment 'CC', Butzin et al. also explored effects of some other model setups, such as the absence of fractionation during photosynthesis ('NP') and more efficient versions of the radiocarbon code ('IC' and 'DA'). They thereby introduce new modelling approaches of radiocarbon, addressing one of the major issues with ($^{14}$)C isotope modelling – the computational cost.

*We thank Anne Morée for her constructive and detailed review and are happy to clarify the issues raised. Corresponding manuscript changes are highlighted in blue in the revision.*

The most important points I would like to raise are as follows.

1. The definitions of the $δ^{13}$C/$δ^{14}$C raise some questions such as which standards are followed (PDB/VPDB), and which constants are used (following OMIP or not?). See detailed comments on L44-50.

*The model uses standard ratios of $^{13}R_{std}$ = $^{14}R_{std}$ = 1. See our reply to your detailed comments on L44-50 further below.*

2. A direct comparison is made with Eide et al. (2017) their PI $δ^{13}$C dataset. This dataset is the result of a subtraction of an estimate of the Suess effect from observational data. It has been shown that the Suess effect is likely underestimated by Eide et al. (2017) in Liu et al. (2021). I think it is important that in the comparison this underestimation is considered and discussed.

*Agreed and done, this is now mentioned and discussed at L233-239.*

3. The $δ^{13}C_{BIO}$ calculation is based on observational data, whereas a model-based separation of $δ^{13}C_{BIO}$ and $δ^{13}C_{AS}$ would be internally consistent and just as easy to calculate. See detailed comments on L222.

*We disagree, see our reply to your detailed comments on L222 further below.*

I would recommend publication of this manuscript after addressing these and the following minor points.

L 37-38: 'numerous ocean general circulation models have been equipped with carbon isotopes and applied in Earth system modelling studies' marine biogeochemistry models have been equipped with C isotopes, right? And only part of your references is for applications in actual ESMs? I think this is an opportunity to highlight it is still quite unique to have C isotopes in an ESM.

*We slightly rephrased the sentence (L37-38).*

L42: $^{12}$C is also a C isotope, so the use of the word 'both' is inaccurate here.

*We added '$^{13}$C and $^{14}$C' in L42.*

L44-50: I think it is relevant to state which standard you use (Pee Dee Belemnite for $\delta^{13}$C), as the new standard is actually the VPDB (Vienna Pee Dee Belemnite) although this is usually not implemented by ESMs because many paleorecords are still reported in the PDB standard.

*Agreed and done, in L80 we refer to the value of most recent reference value of the VPDB standard determined by Assanov et al. (2020). Note that our approach does not refer to a specific standard but considers scaled $^{13}$C and $^{14}$C concentrations implying that our standard values are $^{13}R_{std}$ = $^{14}R_{std}$ = 1. This is clarified in the revision in L83-87.*

L44-50: For $\delta^{14}$C, the CMIP6 standard is $1.170e^{-12}$ and Karlén et al. (1964) is outdated (Orr et al., 2017, see page 2194). Please check in general whether you have followed CMIP6 guidelines, and state in the article if you have deviated from it (this is already done several places but may need to be extended).

*Done, the outdated value has been replaced in L80.*

L44-50: Add a promille symbol after all '1000' in Eq1-3.

*Done, see L47-48, and L50.*

L44-50: How is $^{12}$C calculated (total modelled C minus $^{13}$C?)?

*We approximate total C with $^{12}$C, similar to other models (e.g., Schmittner et al.,2013; Jahn et al., 2015; Liu et al., 2021). See L81 in the revision.*

L63: a sediment model is included, are the C isotopes also in there? If so, how were they initialized? And how is the drift at the end of the control simulation?

*The model version presented here does not include a full-fledged sediment model but a simple one-layer box where impinging detritus is completely remineralized within hours and days. So there is no build-up of sedimentary material. Carbon isotopes are included. The initial concentrations in the layer are close to zero. This is clarified in the revision in L64-65 and L188.*

L77-79: Are they passive tracers? If $^{13}C$ and $^{14}C$ are included in sinking of organic matter, they are not passive in my point of view. Could you specify all carbon compounds which you have included the C isotopes in (e.g., POC, DOC, DIC, PIC, $CaCO_3$, phytoplankton C, zooplankton C?)

*They are passive tracers according to the conventional definition that they do not affect the equation of state (different to temperature and salinity). The model considers carbon isotopes for DIC, DOC, phytoplankton, diatoms, detritus, calcite (of phytoplankton and of detritus), and zooplankton, which is can be found in the revision in L78-79.*

L111: 0.014 should be 0.0144 (Orr et al., 2017)

*We employ 0.0140 following Zhang et al., 1995 (cf. Abstract and Table 4); the true value is already uncertain at the first non-zero decimal (0.0144 ± 0.01; cf. Zhang et al., 1995, main text).*

L119-121: You could also refer here to e.g., Liu et al. (2021) and some other studied as they have explored the difference between these different formulations in MPI-ESM.

*In these lines (now L124-125), we refer to modelling of fractionation processes at the organism level. Global ocean modelling studies considering different parametrizations of $\alpha_P$ (such as the one by Liu et al., 2021) are mentioned further below (L134-135 and L275).*

L124: what makes it 'robust'?

*'Robust' means that is does not rely on species specific assumptions such as species geometry (e.g., shape and size) and species composition. This has been clarified in L128-129.*

L138: In Craig et al. (1954, page 133) it states 'The fractionation factors for $^{14}C$ will then be the square of the $^{13}C$ factors, and, since these numbers are close to 1, the enrichment (fractionation-1) of $^{14}C$ in a given compound should be almost exactly twice that of $^{13}C$ in both equilibrium and rate reaction isotopic effects.' Why use the approximation here instead of the square (i.e., $\alpha^{14}C = (\alpha^{13}C)^2$)? This power of 2 is actually uncertain as well (see detailed discussion on the value on this 'fractionation ratio' in Fahrni et al., 2017;).

*Thanks for pointing us to the paper by Fahrni et al. (2017). Note that already Craig et al. (1954) stated that even taking the square would be an approximation. Therefore, and following the advice of Fahrni et al. (2017) against a change of the fractionation ratio, we stick to $^{14}\alpha = 2\ ^{13}\alpha - 1$ (see equation (3) in Fahrni et al. 2017).*

L138-142: Could you elaborate here what the advantages are of and the reasons for specifically doing this set of experiments? The details of the experiments are mostly given in Sections 3.2 and 3.3, maybe bring them up to here? Or bring forward L341-345?

*Done, the motivation for these various approaches is now explained in L144-147.*

L169-170: Which overturning circulation metrics did you look at for the drift: AMOC? Pacific overturning/Drake Passage? What is the remaining drift in both biogeochemistry (particularly also the C isotopes) and physical state after the full 6000 years of simulation?

*We only checked that strength and depth of the AMOC cell had stabilized, which is now clarified in the revision (L180). Temperature and salinity drifts after 1000 years are about -3 $10^{-4}$ K / a and -3 $10^{-6}$ PSU / a, respectively, in the global average. After 6000 simulated years the temperature drift decreased by two orders of magnitude while the global salinity drift declined by about 25 %. The remaining relative inventory drifts of the major biogeochemical tracers are as follows: $\Delta$ Alk / $\Delta t$ ~ 3 × $10^{-8}$ per year; $\Delta$ DI$^{12}$C / $\Delta t$ and $\Delta$ DI$^{13}$C / $\Delta t$ ~ 2 × $10^{-7}$ per year; $\Delta$ DI$^{14}$C / $\Delta t$ ~ 1 × $10^{-6}$ per year; $\Delta$ DIN / $\Delta t$ ~ 3 × $10^{-6}$ per year; $\Delta$ O$_2$ / $\Delta t$ ~ 3 × $10^{-6}$ per year; $\Delta$ Si / $\Delta t$ ~ 5 × $10^{-6}$ per year. In terms of $\delta^{13}$C and $\Delta^{14}$C, the inventory drift of DI$^{12}$C, DI$^{13}$C and DI$^{14}$C translates to $\Delta$ $\delta^{13}$C$_{DIC}$ / $\Delta t$ ~ 2 × $10^{-5}$ ‰ per year and $\Delta$ $\Delta^{14}$C$_{DIC}$ / $\Delta t$ ~ 1 × $10^{-3}$ ‰ per year, respectively. In the revision (L181-183) we summarize that global-mean temperature and salinity drift by ~ $10^{-6}$ K per year and $10^{-6}$ PSU per year, and that biogeochemical tracer inventories drift by less than $10^{-3}$ percent per year (less than $10^{-3}$ permil per year regarding $^{13}$C$_{DIC}$ and $^{14}$C$_{DIC}$).*

L175-177: If forced with atmospheric concentrations, how do you ensure mass balance?

*In the revision (L191-194) we clarify that the CO$_2$ concentration forcing implies that carbon-isotopic mass is only conserved in the atmosphere-ocean system when the marine isotopic inventories have reached a corresponding steady state. In our simulations this is the case after a few thousand years. Note that this is also the case in any other study with prescribed atmospheric $^{iso}$CO$_2$ concentrations (e.g., Schmittner et al., 2013; Jahn et al., 2015; Liu et al., 2021).*

L181-186: Could you report approximate bias magnitude here for these water masses as well and reflect on how such biases compare to other models?

*Please keep in mind that the low-resolution setup is only used to test new code implementations but not intended for scientific production simulations. It appears that our model results have a higher temperature bias range than the FESOM2 simulations by Scholz et al., 2019 (according to their Fig. 14 ~ ±2 °C vs ±4 °C shown in our Fig. A2). The salinity biases have a similar range (~ ±0.8 PSU according to their Fig. 15 and our Fig. A3). However, these numbers are potentially misleading. The simulations by Scholz et al. 2019 were carried out not only at higher horizontal resolution (127000 vs 3140 horizontal surface nodes), but they were also run over a much shorter period (less than 300 years vs 1000 / 6000 years in our study), used different climate forcing (CORE2 vs CORE NYF), and they were compared with different observations (WOA05 vs WOA09). For these reasons we prefer to remain on the qualitative level but added a specific reference to the above mentioned figures in Scholz et al. (2019), see L197.*

L190-193: How does the biogeochemical state otherwise compare to observations? E.g., Apparent Oxygen Utilization or phosphate, which correlate strongly with $\delta^{13}$C?

*As shown below, the model underestimates dissolved inorganic nitrogen (DIN) in the equatorial Pacific but overestimates DIN in other areas, most notably in upwelling regions. The DIN bias is already reflected by the $\delta^{13}C_{BIO}$ bias (which is proportional to the DIN bias) in Fig. 3. Similarly, the model underestimates apparent oxygen utilization (AOU) at 200 m in the low-latitude Pacific and Atlantic as well as in the interior of North Pacific. To some extent the AOU model deficiencies coincide with the $\delta^{13}C_{DIC}$ differences shown in Fig. 2. However, DIC and DI$^{13}$C do not depend on O$_2$ in REcoM. Therefore, we do not think that these figures would improve our understanding if they were included in the revision.*

[Figure]

L195-196: 'meridional sections': The figures do not show sections but zonal means, please provide the longitude range information (or basin mask?) you have used to make these plots and clarify here.

*We replaced "meridional" with "zonal-mean" (L212). The figures do already include ocean names, and from that it should be clear that "zonal-mean" refers to basin-wide averages.*

L199: 'in wide areas', do you mean over a large part of the ocean at 200m depth?

*This is correct and clarified in L218.*

L206-213: The patterns in Fig. 1 look good; if you would subtract the global mean bias from your model (which you could argue for, especially if you have remaining drift), how good is your agreement then?

*There is no improvement because the simulation results are too high and too low at the same time (see Fig. 2).*

L214: In other studies, $\Delta\delta^{13}C_{DIC}$ is used to designate the vertical marine $\delta^{13}C$ gradient (random relatively recent example: …). I think the use of '$\delta^{13}C$ bias' or something similar would prevent confusion.

*Done, we changed "$\Delta\delta^{13}C_{DIC/BIO/AS}$" to "$\delta^{13}C_{DIC/BIO/AS}$ bias" accordingly (L233 and later on).*

L222: Equation 10, which you have taken from Eide et al. (2017) and adjusted to be able to use DIN, has constants based on observational data as described in Eide et al. (2017). When using a model however, you should in my opinion use the full equation by Maier-Reimer et al. (1992) (see also equation 3 in Eide et al. (2017)), in which you can then insert the model specific parameters. These parameters likely deviate quite a bit from your observational-based Equation 10 (see e.g., Morée et al., 2018; text after Eq. 3). When updating this, the comments made in Lines 231-234 should be updated as well.

*We disagree because we consider $\delta^{13}C_{BIO}$ and $\delta^{13}C_{AS}$ by Eide et al. (2017a) as further benchmarks in addition to observed $\delta^{13}C_{DIC}$. This is clarified in the revision at L253-258. A model validation with reconstructed $\delta^{13}C_{BIO}$ should employ the same transfer function as the reconstruction. Otherwise it could happen that the model yields similar $\delta^{13}C_{BIO}$ values but for different (and maybe wrong) reasons. Note that a thorough analysis should be carried out with 'real' $PO_4$ model results but not with 'pseudo' $PO_4$ values inferred from fixed P / N ratios; all the more, as REcoM explicitly aims at overcoming fixed stoichiometric ratios. A thorough analysis should also include modelled photosynthetic fractionation values instead of a constant value of -19 ‰. Apart from these technical issues it is questionable to what extent the approach by Broecker and Maier-Reimer (1992) leads to meaningful results at all; see also the second review by Andreas Schmittner.*

L236: 'biogenic fractionation', you define $^{13}\alpha_p$ before as '$^{13}\alpha_p$ is the isotopic fractionation factor associated with photosynthesis', please be consistent.

*Done. We replaced "biogenic" with "photosynthetic" (L268).*

L226-230: Could you quantify here (e.g., globally or by region if preferable) what percentage of the bias in $\delta^{13}C_{DIC}$ is due to the $\delta^{13}C_{BIO}$ bias, and what from the residual $\delta^{13}C_{AS}$? This would really highlight where further attention is most needed to reduce mean bias. You could then also add this to the summary at L332.

*This is not possible. An obvious approach would be to consider the bias ratios $\delta\delta^{13}C_{BIO}/\delta\delta^{13}C_{DIC}$ and $\delta\delta^{13}C_{AS}/\delta\delta^{13}C_{DIC}$ but it turns out that the results do not make much sense. While*

$$\delta\delta^{13}C_{BIO}/\delta\delta^{13}C_{DIC} + \delta\delta^{13}C_{AS}/\delta\delta^{13}C_{DIC} = 1,$$

*the individual terms are not normalized, i.e., $|\delta\delta^{13}C_{BIO}/\delta\delta^{13}C_{DIC}| > 1$ and $|\delta\delta^{13}C_{AS}/\delta\delta^{13}C_{DIC}| > 1$ in some areas (see the figure below). This is because (i) $\delta^{13}C_{BIO}$ is derived from phosphorus which (at least in the model) is independent from $\delta^{13}C_{DIC}$, and (ii) $\delta^{13}C_{AS}$ depends on $\delta^{13}C_{BIO}$ at last.*

[Figure]

δδ¹³C_BIO / δδ¹³C_DIC at 200 m  δδ¹³C_AS / δδ¹³C_DIC at 200 m

L261-262: 'This is superimposed by a southward gradient of $\Delta^{14}C_{DIC}$. The meridional $\Delta^{14}C_{DIC}$ gradient reverses in the Pacific.' I do not really follow this. Specify direction of gradient (negative toward south in Atlantic). What reversal do you see in the Pacific?

*In the interior of the oceans $\Delta^{14}C_{DIC}$ decreases from N to S in the Atlantic but from S to N in the Pacific. This has been rephrased accordingly in L294-295.*

L264: what maximum water mass radiocarbon age does that represent? Is your model relatively slowly overturning and therefore relatively old compared to observations (how much older in N-Pacific in terms of e.g. ideal age/radiocarbon age?)? You mention AMOC in L272-274, but formation rates and export of southern source waters would be relevant for maximum water mass age as well outside the north Atlantic.

*The $\Delta^{14}C$ value of -290 ‰ corresponds to a $^{14}C$ age of about 2800 years (depending on the value of the chosen half-life; note that the protocol for $^{14}C$ dating uses a half-life of 5568 years instead of 5700 years). At 3 km depth, the $\Delta^{14}C$ values between the North Atlantic and North Pacific (both considered at 30°N) translate to a radiocarbon age difference of about 1700 years. In the Pacific the northward flux of southern-sourced deep water across 30°N is about 1 Sv, see the figure below. In the revision we now mention that -290 ‰ correspond to about 2800 years (L297-298) and that the MOC is particularly sluggish in the North Pacific (L307).*

[Figure]

L265-270: Could you also here (and possibly at several points in this section) report the bias in terms of radiocarbon age, which may be more intuitive to understand for some readers (and quite comparable to ideal age tracers which almost all models have)?

*We deliberately omit [14]C ages in the paper because such values are not provided by GLODAP. Moreover, a comparison with other tracer ages is not straightforward due to the considerable [14]C age of preformed DI[14]C (e.g., see the discussion by Koeve et al., 2015; doi:10.5194/gmd-8-2079-2015). Numerical age tracers are not implemented in this FESOM version.*

L301: What is a radioconservative tracer? You need $\delta^{13}C$ for the calculation of $\Delta^{14}C$, how do you go about that? I have not understood this experiment based on the description here.

*Fiadeiro (1982) estimated that the effects of biological activity and isotopic fractionation on $^{14}R_{DIC}$ are much smaller than the effects of ocean circulation, mixing and radioactive decay. From that he concluded that $^{14}R_{DIC}$ could be approximately considered as a prognostic and purely physical tracer which is conservative except for its radioactive decay. This approach disregards the marine carbon cycle to the greatest extent, which is clarified in the revision (L336-337). It has been applied in numerous ocean ventilation modelling studies. However, the accuracy of the Delta approximation (DA) in OGCMs has never been checked except for the study by Mouchet (2013).*

L323: 'the correct DI[14]C implementation', do you mean your CC experiment?

*This is correct. We added "(CC)" to the sentence (L359).*

L330-332: I think this sentence does not really summarize your biases. More than the low simulated (CC) $\delta^{13}C$ in upwelling zones, I think the $\delta^{13}C$ and radiocarbon biases are summarized by generally too steep vertical gradients (which leads to upwelling of too-depleted waters for $\delta^{13}C$) as well as too depleted waters at the 'end' of the overturning circulation (as your model overturns relatively slowly). This comment also applied to lines 14-16.

*We agree that the vertical gradients of $\delta^{13}C$ and $\Delta^{14}C_{DIC}$ are too steep in the North Atlantic and have rephrased this accordingly (L367-368). However, this is not really the case for $\delta^{13}C$ in the North Pacific where one could even argue that the vertical gradient of $\delta^{13}C$ weakens.*

L345-346: I think the bias introduced by using the simplified approaches for modelling radiocarbon should be discussed not just relative to experiment CC but also relative to the PI data: I.e., from Fig. 10 it is visible that the already existing bias (too steep gradients) gets even stronger in the simplified approaches.

*Done, in the revision (L381-386) we discuss that the IC approach yields lower $\Delta^{14}C_{DIC}$ values than reconstructed for high latitude surface waters and for deep and bottom waters, while the DA approach yields higher $\Delta^{14}C_{DIC}$ values than reconstructed for surface water which mitigates the isotopic depletion in the deep sea, where this approach therefore better agrees with the reconstruction than the other approaches.*

L619: Here you specify which model experiment you have used (CC), can you do so for all Figs. (e.g., Fig. 1)?

*Done, we added the missing information to the captions of Figs. 1 (L671) and 6 (L697).*

L 688, 694, 699, etc.: If the figure considers WOA data, please specify and cite which WOA data you have used. For all zonal means, please specify over which longitudes the zonal means were taken or whether e.g., some basin mask was used. Instead of showing model and observational data side-to-side, I think it easier to see the differences by showing the model-observation bias like you do in Figs. 2 and 3 (and if you wish to also show the absolute values, keep the model plots too).

*The captions of Figs. A2, A3, and A5 have been updated accordingly (L745, L751, and L763). The figures do already include ocean names, and from that it should be obvious that "zonal-mean" refers to basin-wide averages.*